# Enhancing Rating-Based Reinforcement Learning to Effectively Leverage Feedback from Large Vision-Language Models

Tung M. Luu [1]  Younghwan Lee [1]  Donghoon Lee [1]  Sunho Kim [1]  Min Jun Kim [1]  Chang D. Yoo [1]

## Abstract

Designing effective reward functions remains a fundamental challenge in reinforcement learning (RL), as it often requires extensive human effort and domain expertise. While RL from human feedback has been successful in aligning agents with human intent, acquiring high-quality feedback is costly and labor-intensive, limiting its scalability. Recent advancements in foundation models present a promising alternative–leveraging AI-generated feedback to reduce reliance on human supervision in reward learning. Building on this paradigm, we introduce ERL-VLM, an enhanced rating-based RL method that effectively learns reward functions from AI feedback. Unlike prior methods that rely on pairwise comparisons, ERL-VLM queries large vision-language models (VLMs) for absolute ratings of individual trajectories, enabling more expressive feedback and improved sample efficiency. Additionally, we propose key enhancements to rating-based RL, addressing instability issues caused by data imbalance and noisy labels. Through extensive experiments across both low-level and high-level control tasks, we demonstrate that ERL-VLM significantly outperforms existing VLM-based reward generation methods. Our results demonstrate the potential of AI feedback for scaling RL with minimal human intervention, paving the way for more autonomous and efficient reward learning. The code is available at: https://github.com/tunglm2203/erlvlm.

## 1. Introduction

Reinforcement Learning (RL) has achieved remarkable success in various domains, including games (Silver et al., 2017;

[1]Korea Advanced Institute of Science and Technology (KAIST). Correspondence to: Chang D. Yoo <cd_yoo@kaist.ac.kr>.

*Proceedings of the $42^{nd}$ International Conference on Machine Learning*, Vancouver, Canada. PMLR 267, 2025. Copyright 2025 by the author(s).

Vinyals et al., 2019; Perolat et al., 2022), robotics (Kalashnikov et al., 2018; Chen et al., 2022), and autonomous systems (Bellemare et al., 2020; Zhou et al., 2020; Kaufmann et al., 2023). A critical factor driving these successes is the design of an appropriate reward function, which both captures the task objective and allows an RL agent to learn efficiently. This reward function is often manually defined and refined by human domain experts, a process that is not only time-consuming but also susceptible to errors (Skalse et al., 2022). In this work, we aim to eliminate the dependence on manually designed reward functions by training agents with reward functions derived from data. In this context, RL from human feedback (RLHF) has emerged as a powerful paradigm, enabling the learning of reward functions directly from human feedback (Knox & Stone, 2009; Christiano et al., 2017; Ibarz et al., 2018).

One obstacle for employing RLHF at scale is its reliance on a substantial amount of human feedback, which is both costly and labor-intensive. Recent advancements in vision-language models (VLMs) have demonstrated their ability to understand intricate relationships between images and text. When integrated with large language models, large VLMs have shown a high degree of alignment with human judgment (Gilardi et al., 2023; Ding et al., 2023; Bao et al., 2024; Yoon et al., 2025). These capabilities position large VLMs as promising proxies for human feedback in training RL agents.

Recent studies have explored this approach for training robotic tasks by querying large VLMs for pairwise preferences between trajectories (Guan et al., 2024; Wang et al., 2024a). However, querying VLMs for preferences has a few disadvantages. First, individual preferences convey little information, requiring a large number of queries to learn meaningful reward functions, which can lead to sample inefficiency. In addition, since preferences are inherently derived by comparing at least two trajectories, the number of input and (potential) output tokens processed by the VLM effectively doubles. This leads to higher computational costs and a slower querying process, especially for long trajectories. Finally, when querying VLMs with trajectories of similar quality (*e.g.*, negative-negative or positive-positive pairs), the model often provides invalid preferences by "hal-

lucinating" that one trajectory is better than the other (Guan et al., 2024). To address this, (Wang et al., 2024a) introduces an "unsure" label for indistinguishable trajectories and discards them during reward learning. While this mitigates query ambiguity, it is inefficient as it prevents the full utilization of all queried samples.

To address these challenges, we propose ERL-VLM (**E**nhancing **R**ating-based **L**earning to Effectively Leverage Feedback from **V**ision-**L**anguage **M**odels), a method that efficiently utilizes feedback from large VLMs, such as Gemini (Reid et al., 2024), to generate reward functions for training RL agents. Unlike prior approaches that rely on pairwise comparisons, ERL-VLM queries large VLMs for absolute evaluations of individual trajectories, represented on a Likert scale (*i.e.*, a range of discrete ratings such as "very bad", "bad", "ok", "good", and "very good"). This approach allows VLMs to provide more expressive feedback, reduces query ambiguity, and ensures that all queried samples are fully utilized for reward learning. Additionally, we introduce simple yet effective enhancements to existing rating-based RL methods to mitigate instability in reward learning caused by data imbalance and noisy rating labels from VLMs. We demonstrate that ERL-VLM can generate reward functions that successfully train RL agents across a variety of vision-language navigation discrete action and continuous robotic control environments. Notably, ERL-VLM operates solely on image observations of the agent and a language task description, without requiring privileged state information or access to environment source code, as required in prior work (Ma et al., 2024; Xie et al., 2024; Wang et al., 2024b; Venuto et al., 2024; Han et al., 2024). Our contributions are summarized as follows:

- We present ERL-VLM, a novel method for learning reward functions from VLM-provided feedback. ERL-VLM enables agents to learn new tasks using only a human-provided language task description, significantly reducing the need for extensive human effort in reward design.
- We propose simple yet effective enhancements to existing rating-based RL methods, improving stability in reward learning from VLM feedback and thereby improving the overall agent performance.
- We demonstrate that ERL-VLM outperforms prior VLM-based reward generation methods, successfully training RL agents across a variety of vision-language navigation and robotic control tasks. Additionally, we conduct extensive ablation studies to identify key factors driving ERL-VLM's performance gains and provide deeper insights into its effectiveness.

## 2. Related Work

**Inverse Reinforcement Learning.** Inverse Reinforcement Learning (IRL) aims to recover reward functions from expert demonstrations, enabling the learned reward functions to guide agents in replicating the demonstrated behaviors. Various IRL approaches have been developed, including maximum entropy IRL (Ziebart et al., 2008; Wulfmeier et al., 2015), non-linear IRL (Levine et al., 2011; Finn et al., 2016), adversarial IRL (Ho & Ermon, 2016; Fu et al., 2018), and behavioral cloning IRL (Szot et al., 2023). While these methods have shown success across different domains, they heavily rely on high-quality demonstrations, which are expensive and labor-intensive to obtain. In contrast, ERL-VLM eliminates the need for expert demonstrations by leveraging only human-provided task descriptions in language to learn a reward function.

**Learning from Feedback.** Learning reward functions directly from human feedback has gained significant attention as it typically requires less effort compared to collecting expert demonstrations. Among the most widely used approaches is comparative feedback, where human evaluators perform pairwise comparisons of trajectories to express preferences. These preferences are then transformed into targets for the reward model (Wirth et al., 2017; Christiano et al., 2017; Ibarz et al., 2018; Leike et al., 2018; Lee et al., 2021a;b; Hejna III & Sadigh, 2023). Alternatively, evaluative feedback focuses on providing absolute assessments for individual trajectories (Knox & Stone, 2009; Warnell et al., 2018; MacGlashan et al., 2015; Arumugam et al., 2019; White et al., 2024), potentially offering more comprehensive insights than relative preferences (Casper et al., 2023; Yuan et al., 2024).

While learning from feedback shows promise, it is often constrained by the requirement for extensive human participation to generate the necessary feedback. Recently, learning from AI feedback has emerged as a compelling alternative. Pretrained foundation models have been leveraged to approximate human feedback for RL finetuning in work on large language models (Bai et al., 2022; Li et al., 2023; Lee et al., 2024; Yu et al., 2024). Building on this paradigm, several studies have explored reducing the reliance on human feedback in training RL tasks by utilizing AI-generated preferences (Kwon et al., 2023; Klissarov et al., 2023; Wang et al., 2024a; Lin et al., 2024; Wang et al., 2025). In contrast to these preference-based approaches, our method obtains absolute evaluations instead of relative preferences, which we hypothesize to provide more expressive and informative feedback for RL tasks leveraging AI models.

**Vision-Language Models as Reward Functions.** Recent research has also explored leveraging vision-language models (VLMs), such as Contrastive Language-Image Pretraining (CLIP; (Radford et al., 2021)), to directly generate reward signals for training agents. These rewards are typically computed by measuring the cosine similarity between the embeddings of a language task description and a visual representation of the current state or a sequence of states. For

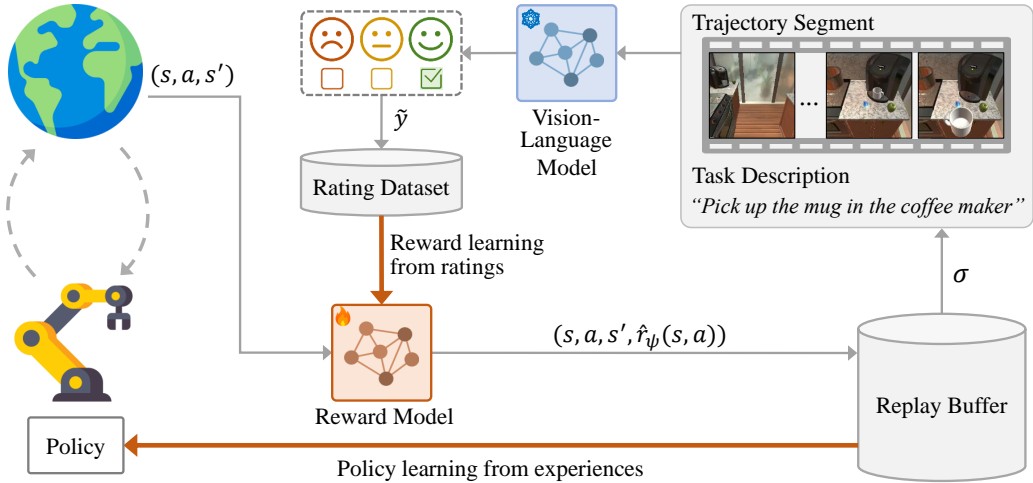

*Figure 1.* **Overview of ERL-VLM**: The figure illustrates an example of a *PickupObject* task in ALFRED along with its corresponding task description. During each feedback session, trajectory segments $\sigma$ are randomly sampled from the replay buffer and sent to the VLM teacher to obtain rating feedback $\tilde{y}$. The reward model $\hat{r}_\psi$ is then trained using the collected rating dataset and subsequently used to compute rewards as the agent interacts with the environment.

example, (Fan et al., 2022; Jiang et al., 2024) train a CLIP model and use it to reward agents in MineCraft environments. However, this approach requires a lot of labeled, environment-specific datasets, which are costly and difficult to scale. In robotics, similar approaches have been used to reward agents, either through fine-tuning or in zero-shot settings (Mahmoudieh et al., 2022; Cui et al., 2022; Baumli et al., 2023; Ma et al., 2023; Rocamonde et al., 2024; Sontakke et al., 2024). Despite their potential in some domains, these similarity score-based reward signals are often noisy, and their accuracy highly dependent on task descriptions and alignment of image observations with the pretraining data distribution (Rocamonde et al., 2024; Sontakke et al., 2024; Wang et al., 2024a). Moreover, most of these methods focus on utilizing the embedding space of VLMs rather than leveraging their reasoning capabilities. In contrast, our approach harnesses the emergent reasoning capabilities of large VLMs, such as Gemini (Reid et al., 2024), to provide rating feedback, which is then distilled into a reward function for training agents. The work most similar to ours is RL-VLM-F (Wang et al., 2024a), which generates preference feedback using VLMs. However, querying VLMs for preferences introduces the risk of query ambiguity. Also, prompting VLMs with multiple trajectories can be computationally inefficient. ERL-VLM instead queries VLMs for absolute ratings for individual trajectories, that improves computational efficiency and ensures all queried samples are fully utilized in reward learning, thereby enhancing the overall performance of training RL agents.

## 3. Preliminaries

In standard reinforcement learning (RL), an agent interacts with an environment in discrete time steps (Sutton & Barto,

2018). At each time step $t$, the agent observes the current state $s_t$ and selects an action $a_t$ according to its policy $\pi(\cdot|s)$, which defines a probability distribution over actions for a given state. Then the environment responses a reward $r_t = r(s_t, a_t)$ and transitions to the next state $s_{t+1}$. The objective of RL algorithms is to find the optimal policy that maximizes the expected return, $\mathcal{R}_0 = \sum_{t=0}^{\infty} \gamma^t r_t$, which is defined as a discounted cumulative sum of the reward with the discount factor $\gamma$.

**Rating-based RL.** In scenarios where a reward function is unavailable, standard RL approaches may not be directly applied to derive policies. RL from human feedback is shown as an effective paradigm for transforming human feedback into guidance signals (Lee et al., 2021b; Casper et al., 2023; Yuan et al., 2024). In this work, we build on the recently introduced rating-based RL framework (White et al., 2024), where a teacher provides discrete ratings for the agent's behaviors (Knox & Stone, 2009; MacGlashan et al., 2017; Arumugam et al., 2019), and an estimated reward function $\hat{r}_\psi$ is trained to ensure that the distribution of predicted ratings matches the distribution of collected ratings. Formally, given a segment $\sigma$ as a sequence of states and actions $\{(s_1, a_1), \dots, (s_H, a_H)\}$, where $H \geq 1$, a teacher assigns a rating label $\tilde{y}$ from the discrete set $\mathcal{C} = \{0, 1, \dots, n-1\}$, to indicate the segment's quality. Here, 0 denotes the lowest possible rating, while $n-1$ represents the highest. Descriptive labels can also be associated with rating levels; for example, with three rating levels, level 0 could be labeled "bad", level 1 as "average", and level 2 as "good". Each feedback is then stored in a dataset $\mathcal{D}$ as a tuple $(\sigma, \tilde{y})$. Given the estimated reward function $\hat{r}_\psi$, the estimated return for a length-$k$ trajectory segment is computed as $\hat{R}(\sigma) = \sum_{t=1}^{k} \hat{r}_\psi(s_t, a_t)$. Then, the probability $P_\sigma(i)$ of assigning

the segment $\sigma$ to the $i$-th rating class is formulated as:

$$P_\sigma(i) = \frac{e^{-(\tilde{R}(\sigma)-\bar{R}_i)(\tilde{R}(\sigma)-\bar{R}_{i+1})}}{\sum_{j=0}^{n-1} e^{-(\tilde{R}(\sigma)-\bar{R}_j)(\tilde{R}(\sigma)-\bar{R}_{j+1})}} \quad (1)$$

where $\tilde{R}(\sigma) \in [0,1]$ denotes the normalized version of $\hat{R}(\sigma)$, computed using min-max normalization within the current training batch. $\bar{R}_i$ represents the rating class boundaries, constrained as $0 := \bar{R}_0 \leq \bar{R}_1 \leq \cdots \leq \bar{R}_n := 1$. These boundaries are determined to ensure that, within the batch, the number of samples in each rating category modeled by Equation (1) reflects the numbers of ratings assigned by the teacher. For details on how the boundaries are computed, we refer readers to (White et al., 2024).

Based on the probabilistic rating model in Equation (1), the estimated reward function $\hat{r}_\psi$ is optimized by minimizing multi-class cross-entropy loss, defined as:

$$\mathcal{L}_{CE}(\psi, \mathcal{D}) = -\mathbb{E}_{(\sigma,\tilde{y})\sim\mathcal{U}(\mathcal{D})}\Big[\sum_{i=0}^{n-1} \mu_\sigma(i) \log P_\sigma(i)\Big] \quad (2)$$

where $\mu_\sigma(i)$ is an indicator function that takes the value 1 if $\tilde{y}$ equals $i$, and 0 otherwise, and $\mathcal{U}$ represents uniform sampling. The policy $\pi$ can then be updated using any RL algorithm to maximize the expected return under the estimated reward function $\hat{r}_\psi$.

**Vision-Language Models for Evaluative Feedback.** In this work, we leverage large pretrained vision-language models (VLMs; (Zhang et al., 2024)) as teachers to provide feedback. We define VLMs as models capable of processing both language inputs, $p = (x_0, \ldots, x_m)$, where $x_m \in \mathcal{V}$, and visual inputs, $I \in \mathcal{I}$. Here, $\mathcal{V}$ represents a finite vocabulary, and $\mathcal{I}$ denotes the space of RGB images. Given these inputs, a VLM $\mathcal{T}$ generates language outputs as $h = \mathcal{T}(p, I)$, where $h = (h_0, \ldots, h_k)$ and $h_k \in \mathcal{V}$. Our focus is on VLMs trained on diverse text and image datasets, which equip them with the ability to generalize effectively across a variety of environments. Additionally, these models must possess strong visual comprehension and question-answering capabilities, critical requirements for accurately evaluating and rating trajectories. In this work, we use Gemini (Reid et al., 2024) as the large pretrained VLM that meets these requirements.

## 4. Method

In this section, we introduce ERL-VLM, a method for learning reward functions from VLM feedback. We first describe the prompt design used to elicit rating feedback from VLMs for the agent's behaviors. Next, as the core contribution of this paper, we introduce enhancements to the existing rating-based RL framework to effectively utilize this feedback in learning the reward function. Finally, we outline how the learned reward function is used for policy learning. An overview of ERL-VLM is presented in Figure 1, and the

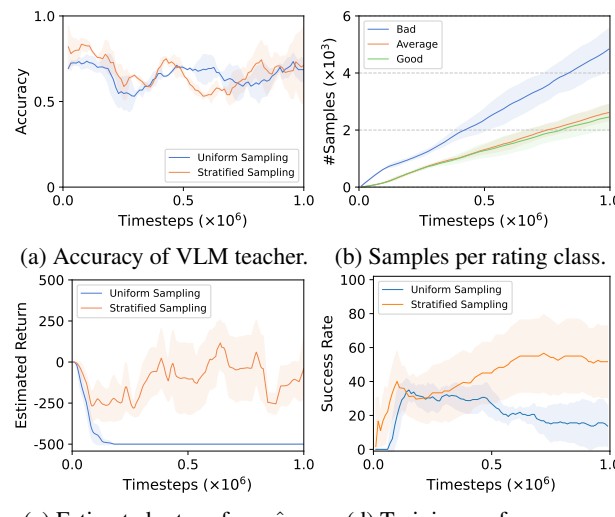

(a) Accuracy of VLM teacher.  (b) Samples per rating class.

(c) Estimated return from $\hat{r}_\psi$.  (d) Training performance.

*Figure 2.* Reward learning from rating feedback in the *Drawer Open* environment. Due to the imbalance of ratings, optimizing $\hat{r}_\psi$ using the conventional objective in Eq. (2) with uniform batch sampling often leads to poor performance. To mitigate this issue, we use stratified sampling to ensure that every training batch contains samples from all rating classes.

complete procedure is detailed in Algorithm 1.

**Rating Generation with VLMs.** In this work, we focus on tasks where the quality or success of an agent's behaviors can be evaluated based on a single image or a sequence of images. Building on prior work (Wang et al., 2024a), we elicit feedback from the VLM through a two-stage process: first analyzing the agent's behavior and then generating ratings based on this analysis. We observe that the quality of ratings heavily depends on the effectiveness of the analysis stage. For tasks with simple descriptions, the VLM is capable of producing plausible ratings from a single image observation. However, for tasks with complex and compositional descriptions, single observations often fail to capture the full context. To address this, we design a prompt that directs the VLM to analyze trajectory segments composed of multiple image observations and, if applicable, associated actions. This approach provides greater granularity, enabling a more comprehensive analysis that captures critical nuances in the agent's behavior. The prompt templates used in our experiments are shown in Table 6 and 7. Given these prompts, during each feedback session, we sample $N$ segments from the replay buffer and query the VLM to rate the segments according to the task description. The sampled segment, along with its corresponding rating, is then stored in the rating dataset $\mathcal{D}$ (lines 12-14 in Algorithm 1).

**Reward Learning from Ratings.** Given the rating dataset $\mathcal{D}$ collected from the VLM teacher, the estimated reward function $\hat{r}_\psi$ can be optimized using the conventional objective in Equation (2). However, directly applying this

**Algorithm 1** ERL-VLM

1: **Input**: Text description of task goal $l$.
2: **Initialize**: Policy $\pi_\theta$ and reward function $\hat{r}_\psi$, RL replay buffer $\mathcal{B} \leftarrow \emptyset$, dataset of rating $\mathcal{D} \leftarrow \emptyset$, VLM query frequency $K$, number of queries $N$ per feedback session, horizon $T$.
3: **for** each iteration **do**
4:    // DATA COLLECTION
5:    **for** $t = 1$ to $T$ **do**
6:       Collect $s_{t+1}$, image $I_{t+1}$ by taking $a_t \sim \pi_\theta(a_t|s_t)$
7:       Update $\mathcal{B} \leftarrow \mathcal{B} \cup \{(s_t, I_t, a_t, s_{t+1}, I_{t+1}, \hat{r}_t)\}$
8:    **end for**
9:    // RATING BY VLM AND REWARD LEARNING
10:   **if** iteration $\% K == 0$ **then**
11:     **for** $m = 1$ to $N$ **do**
12:       Randomly sample a segment $\sigma$ from $\mathcal{B}$
13:       Query VLM with $\sigma$ and task goal $l$ for label $\tilde{y}$
14:       Update $\mathcal{D} \leftarrow \mathcal{D} \cup \{(\sigma, \tilde{y})\}$
15:     **end for**
16:     **for** each gradient step **do**
17:       Stratified sampling minibatch $\{(\sigma, \tilde{y})_j\}_{j=1}^D \sim \mathcal{D}$
18:       Update $\hat{r}_\psi$ according to Equation (3)
19:     **end for**
20:     Relabel entire replay buffer $\mathcal{B}$ using updated $\hat{r}_\psi$
21:   **end if**
22:   // POLICY LEARNING
23:   **for** each gradient step **do**
24:     Sample random batch $\{(s_t, a_t, s_{t+1}, \hat{r}_t)_j\}_{j=1}^B \sim \mathcal{B}$
25:     Update the policy $\pi_\theta$ with any off-policy RL algorithm
26:   **end for**
27: **end for**

approach often leads to instability during reward learning, even when the teacher provides reasonable ratings, as shown in Figure 2a. This instability arises from two key challenges.

First, the distribution of samples across rating classes is often highly imbalanced. For example, in the early stages of training, "bad" ratings dominate other rating classes, resulting in skewed datasets (Figure 2b). This imbalance negatively impacts reward learning because multiple training batches may contain samples from only a single rating class, which disrupts the effective selection of rating class boundaries. Consequently, the reward function tends to consistently predict the dominant rating class, leading to degraded performance, as shown in Figure 2c, 2d. It is worth noting that this issue is less prevalent in preference-based RL, as the binary comparisons inherently balance the preferences between two samples. To address this issue, we employ stratified sampling, ensuring that samples from all rating classes are represented in each training batch. Additionally, we use a weighted loss function, assigning weights based on class frequency to account for data imbalance.

Second, the VLM teacher may provide noisy or imprecise labels due to hallucinations (Zhang et al., 2023b) or other errors. To mitigate the impact of noisy labels, we adopt the Mean Absolute Error (MAE) as the training objective, as it is provably more robust to label noise compared to Cross

Entropy (Ghosh et al., 2017). The reward function $\hat{r}_\psi$ is then optimized using the following objective:

$$\mathcal{L}_{MAE}(\psi, \mathcal{D}) = \mathbb{E}_{(\sigma, \tilde{y}) \sim \mathcal{U}_S(\mathcal{D})}\left[\sum_{i=0}^{n-1} |\mu_\sigma(i) - P_\sigma(i)|\right] \quad (3)$$

where, $\mathcal{U}_S$ denotes the stratified sampling strategy. Prior works in preference-based RL often adopt label smoothing (Wei et al., 2021; Christiano et al., 2017; Ibarz et al., 2018) to handle corrupted labels, assuming a fixed rate of uniform noise in the preferences provided by the teacher. However, in our experiments, we find this approach to be ineffective, likely because it is challenging to estimate the percentage of corrupted labels produced by the VLM teacher, especially in the context of multi-class ratings.

**Implementation Details.** Following prior works in RLHF (Lee et al., 2021a;b; White et al., 2024), we query the VLM teacher every $K$ training steps (Algorithm 1 lines 11-15). At the end of each feedback session, we optimize the reward function $\hat{r}_\psi$ using the objective defined in Equation 3 (Algorithm 1 lines 17-18). Subsequently, all experiences in the replay buffer are relabeled with the newly updated reward function. Finally, the policy is trained using an RL algorithm with the relabeled data from the replay buffer. In this work, we adopt SAC (Haarnoja et al., 2018) and a variant of IQL (Kostrikov et al., 2022) as the base RL algorithms.

## 5. Experiments

The goal of our experiments is to evaluate ERL-VLM's effectiveness in generating reward functions for training RL agents. We compare ERL-VLM to VLM-based reward generation methods, which also operate on image observations and language task description, across three domains: low-level manipulation control tasks in MetaWorld (Yu et al., 2020), high-level vision-language navigation tasks in AL-FRED (Shridhar et al., 2020), and real-world robotic manipulation using a Sawyer robotic arm. Concretely, we aim to answer the following questions: (1) Can ERL-VLM generate effective reward functions for policy learning? (2) How do reward functions generated by ERL-VLM compare to those produced by other pretrained VLMs? (3) What is the impact of our proposed enhancements on the overall effectiveness of ERL-VLM?

### 5.1. Experimental Setup

**MetaWorld.** In this environment, the agent generates low-level continuous actions to control a simulated Sawyer robot, enabling it to interact with objects on a table to complete tasks. We evaluate ERL-VLM on three tasks, following prior work (Sontakke et al., 2024; Wang et al., 2024a):

- *Sweep Into*: Sweep a green puck into a hole on the table.
- *Drawer Open*: Pull open a drawer on the table.

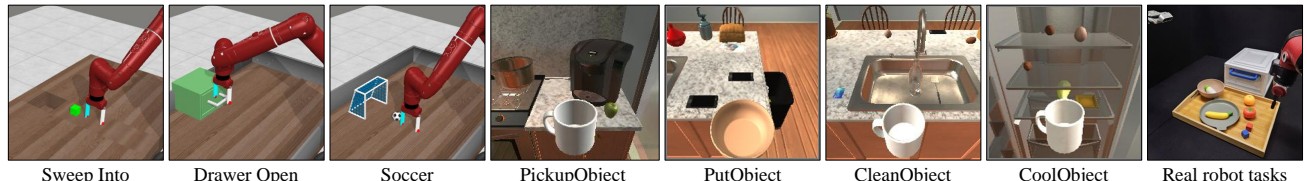

| Sweep Into | Drawer Open | Soccer | PickupObject | PutObject | CleanObject | CoolObject | Real robot tasks |

*Figure 3.* We evaluate ERL-VLM across three domains: two simulated environments and one real-world robotic manipulation setup. The first three images depict tasks from MetaWorld, the next four images illustrate examples from ALFRED, and the final image shows the real-world robotic setup.

• *Soccer*: Kick a soccer into the goal on the table.

**ALFRED.** In this domain, an embodied agent is randomly placed in a room and must select appropriate discrete actions to complete tasks specified by language instructions. ALFRED abstracts away low-level control into 12 discrete actions and along with 82 discrete object types. We leverage a modified version of the ALFRED simulator (Zhang et al., 2023a) that enables online RL training. We consider a set of 20 tasks drawn from 10 scenes in the Kitchen environment. These tasks can be categorized into four types:

• *PickupObject*: Pick up a specified object from a location.
• *PutObject*: Place an object into a target location.
• *CoolObject*: Interact with a fridge to cool an object.
• *CleanObject*: Use a faucet to clean an object.

The object types and desired locations are specified in the instructions. We use the original crowd-sourced language instructions from the benchmark as task descriptions. This allows us to test the robustness of the VLM teacher in providing feedback based on various human-provided task descriptions. Detailed task instructions are provided in Table 2. Unlike MetaWorld, this domain requires multi-task learning, where the policy is conditioned on a language instruction. At the start of each episode, a scene is randomly sampled from a set of 10 scenes, with objects initialized based on the scene's configuration. Each scene includes two sequential instructions, requiring the agent to complete the first task before proceeding to the next.

The visualizations of these tasks are shown in Figure 3. Further details about the tasks and environment setups can be found in the Appendix.

**Baselines.** We compare ERL-VLM to established methods that also leverage pretrained VLMs to generate rewards for robotic tasks based on the language task description and the agent's image observations. These methods involve (1) using similarity scores as reward signals and (2) utilizing feedback from VLMs to learn reward functions. Concretely, we compare against the following baselines:

• **CLIP Score** (Rocamonde et al., 2024): Generate rewards by computing the cosine similarity between the current image observation and the task description in CLIP embedding space (Radford et al., 2021). This reward gen-

eration approach has also been explored in several prior works (Cui et al., 2022; Mahmoudieh et al., 2022).
• **RoboCLIP Score**: We use the text-based version of Robo-CLIP, where rewards are computed based on the similarity between the video embedding of a trajectory and a task description in S3D embedding space (Xie et al., 2018).
• **RL-VLM-F** (Wang et al., 2024a): Similar to our approach, this baseline leverages a pretrained VLM to provide feedback for reward learning; however, it queries for relative preferences rather than absolute ratings.
• **Environment Reward**: This baseline trains the agent using the reward function from environments. The reward function is dense in MetaWorld but sparse in ALFRED.

**Training and Evaluation Procedure.** For MetaWorld tasks, we adopt Soft Actor-Critic (SAC) (Haarnoja et al., 2018) as the base RL algorithm, while for ALFRED tasks, we use an online variant of Implicit Q-Learning (IQL) (Kostrikov et al., 2022), which has been shown to effectively train agents in ALFRED (Zhang et al., 2023a). To ensure a fair comparison, we use the same hyperparameters for policy training across all methods, with the only difference being the reward function. For methods that require learning reward functions, we use Gemini-1.5-Pro (Reid et al., 2024) as the VLM for all tasks and use same query budget (10000 for MetaWorld and 1500 for ALFRED), and apply a uniform sampling scheme for queries. Additionally, the reward function in ALFRED also conditions in the task instruction, similar to the policy. For RL-VLM-F, we perform pairwise comparisons between trajectories generated from the same instruction in ALFRED. We evaluate performance using task success rate, as defined by each benchmark. Each evaluation run computes success rates over 10 episodes in MetaWorld and 100 episodes in ALFRED. Performance are reported as the mean and standard deviation over 3 runs.

### 5.2. Main Results

We present the learning curves of success rates for all compared methods across two simulated domains in Figure 4. As shown, ERL-VLM outperforms all baselines in 6 out of 7 tasks, demonstrating its effectiveness in learning reward functions from VLM feedback. For similarity score-based approaches, while they provide useful learning signals, their

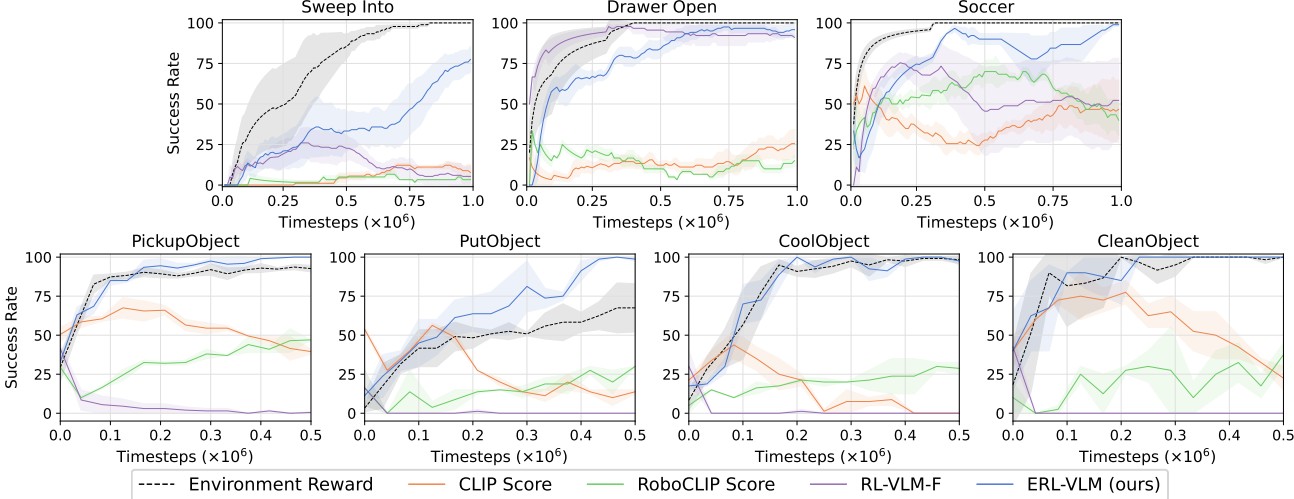

*Figure 4.* Success rate for ERL-VLM and baselines across 3 manipulation tasks from MetaWorld and 20 vision-language navigation tasks from ALFRED. Tasks of the same type in ALFRED are aggregated for clearer visualization. Our method achieves prominent performance on all tasks, and significantly outperforms baselines on ALFRED tasks. Results are means of 3 runs with standard deviation (shaded area).

performance often fluctuates throughout training. This inconsistency arises from the inherent noisiness of the reward signals, as also observed in prior works (Ma et al., 2023; Sontakke et al., 2024; Rocamonde et al., 2024; Wang et al., 2024a). This suggests that relying on direct embedding similarities results in unreliable reward values, leading to instability in policy optimization.

For reward learning from VLM approaches, RL-VLM-F performs comparably to ERL-VLM in *Drawer Open* but struggles significantly in other tasks, particularly in ALFRED, where it nearly fails to learn meaningful policies. We attribute this to the limited query budget, which results in a relatively small number of preference feedback samples per task in ALFRED (*e.g.*, 75 in our experiments). Given that each preference only conveys little information about relative ranking between two segments, the reward function struggles to capture the desired behavioral nuances effectively. In contrast, ERL-VLM leverages absolute ratings, which offer greater global value than preferences within the same query budget. This results in a more informative reward signal for policy training, enabling more effective utilization of VLM feedback.

Interestingly, for the tasks of *PickupObject* and *PutObject* tasks, ERL-VLM outperforms even the sparse reward function. This suggests that learning a reward function from absolute ratings not only provides useful learning signals—similar to sparse rewards—but also introduces more shaping signals that emphasize critical states for task completion. We attribute this to the improved analysis in our prompt design, which enables the VLM to conduct a deeper assessment of agent behaviors. This result highlights the potential of VLMs to serve as effective substitutes for human providing feedback in reward learning.

*Table 1.* Success rates on the 3 real-world robot evaluation tasks in table-top environment.

| Method | Sweep Bowl | Drawer Open | Pickup Banana |
|---|---|---|---|
| BC | $0.50 \pm 0.10$ | $0.23 \pm 0.06$ | $0.17 \pm 0.06$ |
| Sparse Rewards | $0.57 \pm 0.06$ | $0.37 \pm 0.06$ | $0.30 \pm 0.10$ |
| ERL-VLM (Ours) | $\mathbf{0.73 \pm 0.06}$ | $\mathbf{0.60 \pm 0.10}$ | $\mathbf{0.47 \pm 0.12}$ |

### 5.3. Real Robot Evaluation

We further evaluate ERL-VLM in a real-world robotic manipulation setting using a 7-DOF Rethink Sawyer robot interacting with objects in a tabletop environment. We compare our approach against two baseline approaches: (1) *Behavior Cloning (BC)*, a supervised policy pretraining method, and (2) *IQL with Sparse Rewards*, an offline RL setting where the agent learns from sparsely provided rewards. Image observations are captured by a RealSense camera, as illustrated in Figure 3 (rightmost). We consider following tasks:

- *Sweep Bowl*: Sweep a bowl across a red boundary line.
- *Drawer Open*: Pull open a drawer.
- *Pickup Banana*: Grasp a banana from a plate and lift it.

For each task, we collect 50 demonstrations via human teleoperation and query Gemini to obtain rating feedback offline. The reward model is then trained using the objective in Equation 3. The learned reward function is subsequently used to relabel rewards in the offline dataset. We employ IQL (Kostrikov et al., 2022) to train the policy in an offline manner, following the setup in (Yuan et al., 2024). Results in Table 1 demonstrate that ERL-VLM is capable of generating useful reward functions that facilitate policy learning, indicating its potential for real-world robot manipulation tasks. More visualization results are shown in Figure 9.

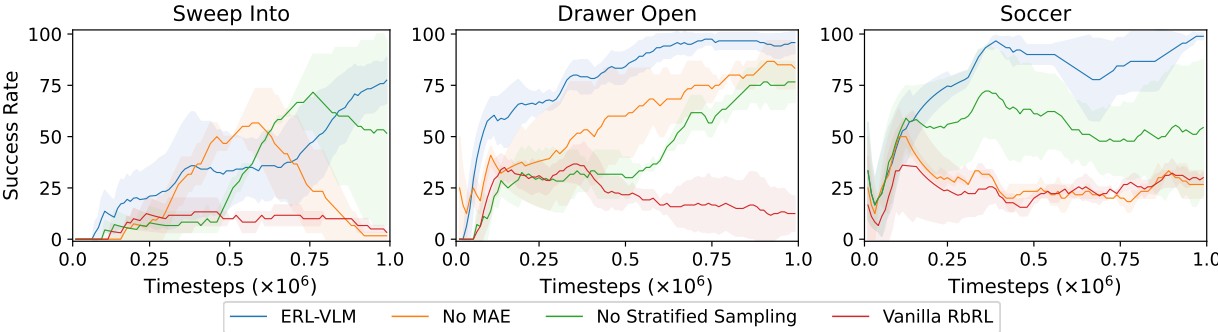

*Figure 5.* Ablation studies with different enhancements in ERL-VLM. Results are means of 3 runs with standard deviation (shaded area).

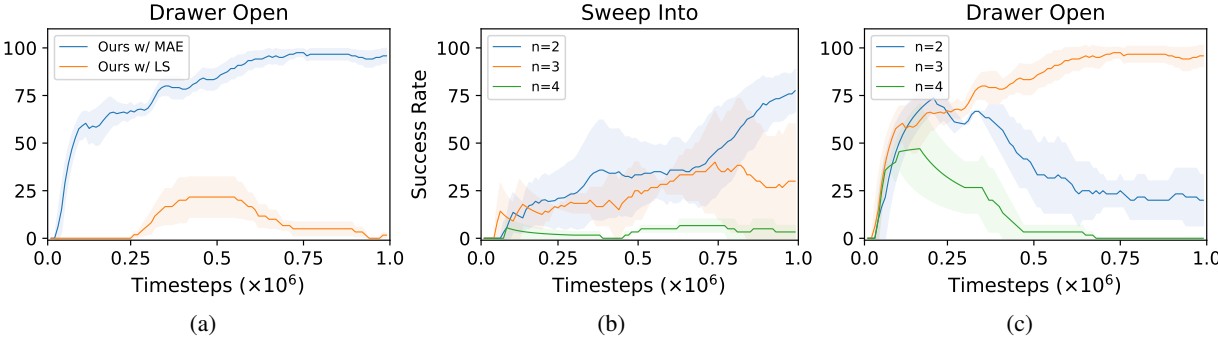

*Figure 6.* (a) The effect of label smoothing in handling the corrupted labels. (b/c) ERL-VLM with different numbers of rating classes.

### 5.4. Ablation Studies

**Enhancements to Rating-based RL.** To better analyze the impact of our core contributions, we conduct ablation studies on MetaWorld tasks using the following variants:

- **Vanilla RbRL**: The original rating-based RL framework (White et al., 2024), which optimizes the reward function using cross-entropy loss with uniform sampling, as defined in Equation (2).
- **No MAE**: A variant of ERL-VLM that replaces mean absolute error (MAE) loss with cross-entropy loss while retaining stratified sampling and weighted loss, mirroring Vanilla RbRL with data balancing improvements.
- **No Stratified Sampling**: A variant of ERL-VLM that removes stratified sampling and weighted loss, leaving only MAE as the enhancement over Vanilla RbRL.

We report results in Figure 5. The analysis shows that each enhancement contributes to the performance of ERL-VLM when learning from rating feedback. Notably, adopting MAE loss provides the most significant improvement over Vanilla RbRL, demonstrating its robustness in handling noisy labels generated by VLMs. Additionally, stratified sampling and weighted loss effectively mitigate data imbalance, particularly in *Sweep Into* and *Drawer Open*. We further investigate the impact of label smoothing (LS) (Wei et al., 2021) as an alternative strategy for handling corrupted labels in *Drawer Open* (3 rating levels). Specifically, we replace the MAE loss with cross-entropy loss using a soft label: $(1 - r) \cdot \tilde{y} + r/3 \cdot [1, 1, 1]^\top$, where $r = 0.1$, following prior work (Ibarz et al., 2018; Lee et al., 2021b). The results, shown in Figure 6 (a), indicate that label smoothing is ineffective in our experiment.

**Different Number of Rating Classes.** To evaluate the impact of the number of rating classes $n$, we modify the prompt to query VLM with $n = \{2, 3, 4\}$ and evaluate performance on *Sweep Into* and *Drawer Open*. The results, shown in Figure 6 (b/c), reveal that increasing $n$ (*e.g.*, $n = 4$) degrades performance in both tasks, likely due to increased ambiguity and inconsistency in VLM-generated ratings. Interestingly, while $n = 2$ performs best in *Sweep Into*, $n = 3$ yields superior results in *Drawer Open*. We attribute this to task-specific differences in evaluation clarity: in *Sweep Into*, evaluating whether the puck has placed in the hole is binary and unambiguous, whereas in *Drawer Open*, the degree of opening is more subjective, making a three-class rating scheme more suitable. We anticipate that more refined prompt engineering, with carefully crafted rating descriptions, could further exploit the potential of VLMs for feedback.

## 6. Conclusion

In this work, we propose ERL-VLM, a method for learning reward functions by querying VLMs for rating feedback based on a human-provided language task description and

image observations. Unlike preference-based approaches, ours leverages VLM feedback more effectively and efficiently by querying for absolute evaluations rather than pairwise comparisons. We demonstrate its effectiveness across low-level continuous control, high-level vision-language navigation, and real-world robotic manipulation tasks. Our results highlight ERL-VLM's potential to reduce human effort in reward design, paving the way for more scalable and autonomous RL systems.

## Impact Statement

Our approach offers a promising approach to aligning RL agents with more fine-grained human intent by leveraging vision-language models (VLMs) for reward learning. By carefully designing natural language prompts, our method enables rapid and scalable feedback collection, significantly reducing human effort in prototyping and deploying RL systems in real-world applications. However, as our method relies on large foundation models, it inherits their inherent biases, which could propagate into RL agents. This raises important considerations regarding robustness and safety, especially in safety-critical applications. Ensuring the reliability of AI-driven feedback remains a crucial challenge. In this work, we use only visual observations and descriptive actions to analyze agent behavior. Future research can explore integrating additional sensing modalities to further enhance the accuracy and reliability of feedback from foundation models, ultimately leading to more trustworthy and effective RL systems.

## Acknowledgements

This work was supported by Institute for Information & communications Technology Planning & Evaluation (IITP) grant funded by the Korea government(MSIT) (No.RS-2021-II211381, Development of Causal AI through Video Understanding and Reinforcement Learning, and Its Applications to Real Environments) and partly supported by Institute of Information & communications Technology Planning & Evaluation (IITP) grant funded by the Korea government(MSIT) (No.RS-2022-II220184, Development and Study of AI Technologies to Inexpensively Conform to Evolving Policy on Ethics).

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

# Appendix

# A. Details on Tasks and Environments

## A.1. MetaWorld Tasks

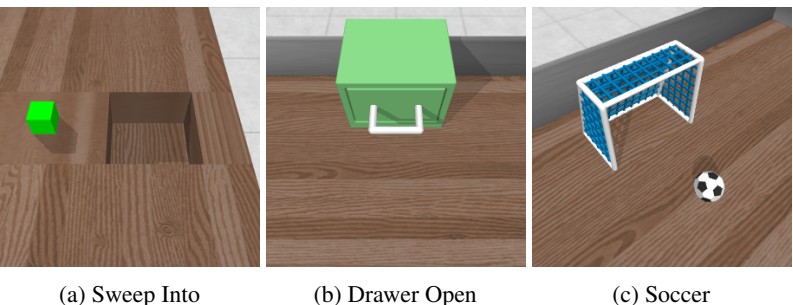

(a) Sweep Into       (b) Drawer Open       (c) Soccer

*Figure 7.* Examples of image observations in MetaWorld for querying the VLM teacher and reward learning.

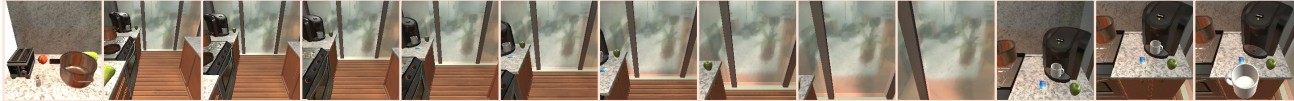

*Figure 8.* Example of trajectory segment from a "PickupObject" task in scene 4.

We evaluate our methods on three tasks from MetaWorld (Yu et al., 2020), following prior work (Sontakke et al., 2024; Wang et al., 2024a). We describe tasks as follows:

- **Sweep Into**: The goal is to sweep a green puck into a square hole on the table. The task is considered successful when the puck remains inside the hole.
- **Drawer Open**: The goal is to pull open a drawer on the table. The task is considered successful when the drawer extends beyond a certain threshold.
- **Soccer**: The goal is to kick a soccer ball into the goal. The task is considered successful when the ball remains inside the goal area.

**Observation Space.** We use state-based observations for policy learning, whereas for reward learning, we use RGB image-based observations rendered from the simulator. Following the original MetaWorld setup (Yu et al., 2020), the state observation is a 39-dimensional vector consisting of the end-effector position, the gripper status, the object's position, the object's orientation, and the goal position. For image observations, we render RGB images at a resolution of $224 \times 224$. Additionally, as these tasks are all object-centric, we follow (Wang et al., 2024a) to remove the robot from the image to facilitate the analyzing process of VLMs. The examples of image observation are shown in Figure 7.

**Action Space.** The action is a 4-dimensional vector, normalized within $[-1, 1]$. The first three dimensions represent the displacement in the position of the end-effector, and the last dimension is a normalized torque applied to the gripper fingers.

## A.2. ALFRED Tasks

We evaluate our methods in a modified version of the ALFRED simulator (Zhang et al., 2023a) to assess its effectiveness in solving high-level vision-language navigation tasks. We focus on 20 tasks drawn from 10 scenes in the Kitchen environment, which requires both navigation and object manipulation. Each scene includes two sequential instructions, and the agent must complete the first instruction before proceeding to the next. The detailed task instructions are provided in Table 2.

**Observation Space.** For both policy learning and reward learning, we use input observations composed of the $224 \times 224$ egocentric RGB image and the robot's pose. The robot's pose is 4-dimensional vector, consisting of its $(x, y, z)$ coordinates and camera horizon angle. The camera horizon can take values from $\{-45, -30, \dots, 30, 45\}$. For all methods, we preprocess the image observation using a frozen ResNet-18 encoder (He et al., 2016) pretrained on ImageNet, obtaining a visual representation of shape $512 \times 7 \times 7$. The example of image observations are shown in Figure 8.

**Action Space.** In ALFRED, the agent operates with a discrete action space of 12 high-level actions, including 5 navigation actions: MoveAhead, RotateRight, RotateLeft, LookUp, and LookDown, and 7 interaction actions: Put, Pickup, Open, Close, ToggleOn, ToggleOff, and Slice. For interaction actions, the policy additionally selects one of 82 object types to interact with. Note that for the VLM analysis prompt, we use only actions and not object types. Due to large discrete action space (5 + 7 * 82), we perform same masking as (Zhang et al., 2023a) to prevent agents from taking actions that are not possible (*e.g.*, the policy cannot output Close for object Tomato).

| Scene | Task Type | Task Instruction |
|---|---|---|
| 1 | PickupObject | Pick up the spoon from the counter |
| | PutObject | Put the spoon in the white cup on the shelf. |
| 2 | PickupObject | Pick up the egg that is beside the fork in the sink. |
| | CoolObject | Open the refrigerator, then place the egg on the glass shelf and close the fridge. Wait then open the fridge and pick up the egg, then close the fridge. |
| 3 | PickupObject | Pick up the tomato from the sink. |
| | CoolObject | Open the fridge door, put the tomato inside of the fridge, close the door, open the door, take the tomato out, close the door. |
| 4 | PickupObject | Pick up the mug in the coffee maker |
| | CoolObject | Open the fridge, put the cup in the fridge, close the fridge, wait, open the fridge, pick the cup, close the fridge |
| 5 | PickupObject | Pick up the bread. |
| | CoolObject | Open the fridge, put the bread in the fridge, close the fridge, open the fridge, get the bread, and close the fridge. |
| 6 | PickupObject | Pick up the white coffee cup to the right of the trophy. |
| | CleanObject | Put the coffee cup in the sink, turn on the water, turn off the water and pick up the coffee cup. |
| 7 | PickupObject | Pick up the smaller silver knife on the counter. |
| | PutObject | Put the knife in the green cup in the sink. |
| 8 | PickupObject | Pick up a bowl from the shelf |
| | PutObject | Put the bowl on the counter |
| 9 | PickupObject | Grab the knife from the counter |
| | PutObject | Put the knife in the pan on the stove |
| 10 | PickupObject | Pick up the knife from the counter. |
| | CleanObject | Place the knife in the sink and turn the water on. Turn the water off and pick up the knife. |

*Table 2.* Task instructions from the Kitchen environment in ALFRED.

### A.3. Real Robot Tasks

**Observation Space.** For policy learning, we use input observations composed of the $480 \times 480$ RGB image captured by an Intel RealSense D435i and the end-effector's state. For all methods, we preprocess the image by first downsampling it to $224 \times 224$ and then transforming it into a 512-dimensional feature vector using a pretrained R3M image encoder (Nair et al., 2022). The end-effector state is a 5-dimensional vector consisting of the end-effector's position $(x, y, z)$, yaw orientation, and gripper's binary state (open/close). We concatenate the image feature with the end-effector state to form the final input observation, resulting in a 517-dimensional vector.

**Action Space.** The action space consists of (1) the displacement of the end-effector's position in Cartesian space and (2) a discrete value indicating whether the gripper should open or close. These actions are transmitted to the robot with 10 Hz and converted to the desired joint poses using the Sawyer robot's Intera SDK inverse kinematics solver.

## B. Training Implementation Details and Hyperparameters

### B.1. MetaWorld

**Policy Training.** We build on RbRL (White et al., 2024) for training policy, which is similar to PEBBLE (Lee et al., 2021a), except that it learns from rating feedback. Original RbRL's implementation[1] only supports locomotion tasks, we extend it to support manipulation tasks in MetaWorld. We use SAC (Haarnoja et al., 2018) as the off-policy RL algorithm, the detailed hyperparameters are shown in Table 3. For all methods, the policy is learned with state-based observations.

---

[1]https://github.com/Dev1nW/Rating-based-Reinforcement-Learning

| Parameter | Value |
|---|---|
| Number of layers | 3 |
| Hidden units per each layer | 256 |
| Steps of unsupervised pre-training | 9000 |
| # Training Steps | $1e6$ |
| Batch Size | 512 |
| Learning rate | 0.0003 |
| Optimizer | Adam |
| Initial temperature | 0.1 |
| Critic target update freq | 2 |
| Critic EMA $\tau$ | 0.005 |
| $(\beta_1, \beta_2)$ | (0.9, 0.999) |
| Discount $\bar{\gamma}$ | 0.99 |
| VLM query frequency $K$ | 5000 |
| Number of queries $N$ per session | 50 |
| Maximum budget of queries | 10000 |

*Table 3.* Hyperparameters for SAC in MetaWorld.

**Reward Learning.** Following (Wang et al., 2024a), we use an image-based reward model consisting of a 4-layer Convolutional Neural Network, followed by a linear layer and a Tanh activation. Additionally, we employ an ensemble of three reward models. For all tasks, we use a segment size of 1 for reward learning. The reward models are optimized using loss function defined in Equation (3). We use batch size of 512 and Adam as the optimizer with the learning rate of 0.0003.

### B.2. ALFRED

**Policy Learning.** We use Implicit Q-Learning (IQL) (Kostrikov et al., 2022) with a transformer-based architecture for both the policy and critic networks, similar to (Zhang et al., 2023a). The hyperparameters are provided in Table 4. The main difference is that we set the quantile parameter $\tau = 0.5$, making IQL a standard off-policy online RL algorithm, rather than one suited for offline RL as in (Zhang et al., 2023a). During training, to reduce exploration time—which is particularly challenging in ALFRED—we seed the buffer with 3 human-collected demonstrations for each task. It is important to note that while these demonstrations complete the task, they are not necessarily optimal. This is applied consistently across all baselines. Additionally, we relabel the rewards in the seed buffer to align with each baseline's reward function, ensuring compatibility during training. For all methods, the policy takes input as a sequence of states and actions as input, and each state is structured as mentioned in Section A.2.

**Reward Learning.** For the reward model, we use the same transformer-based architecture as the policy, which also takes inputs same as the policy. The output of the reward model is bounded by a Tanh activation function. We also employ an ensemble of three reward models, similar to the setup in MetaWorld. Additionally, since the buffer is seeded with demonstrations, we first query the VLM for roughly 1000 feedback samples and perform warmup training on the reward model using these feedback. During online training, we continue training the reward model with remaining query budget. The reward models are optimized using the loss function defined in Equation (3). We use batch size of 512 and AdamW as the optimizer with a learning rate of 0.0001. This setup is applied to both RL-VLM-F and ERL-VLM.

### B.3. Real Robot

**Policy Learning.** We train the policy and critic networks using Implicit Q-Learning (IQL) (Kostrikov et al., 2022), employing a transformer-based architecture similar to ALFRED tasks. The hyperparameters are provided in Table 5. For the Sparse Rewards baseline, we assign a reward of one at the final step of the demonstrations. Since the gripper action space is discrete and the dataset is imbalanced, we adjust the gripper loss by applying inverse frequency weighting based on the number of examples in each class.

**Reward Learning.** The reward model utilizes the same Convolutional Neural Network architecture as MetaWorld environment. We use a Tanh activation function to bound the output. We also employ an ensemble of three reward models.

| Parameter | Value |
|---|---|
| Batch Size | 128 |
| # Training Steps | $5e5$ |
| Learning Rate | 0.0001 |
| Optimizer | AdamW |
| Dropout Rate | 0.1 |
| Weight Decay | 0.1 |
| Discount $\gamma$ | 0.97 |
| Q Update Polyak Averaging Coefficient | 0.005 |
| Policy and Q Update Period | 8 per train iter |
| IQL Advantage Clipping | [0, 100] |
| IQL Advantage Inverse Temperature $\beta$ | 5 |
| *IQL Qunatile $\tau$* | **0.5** |
| Maximum Context Length | 8 |
| Warmup query | $\sim$1000 |
| # Epochs for warmup training reward model | 100 |
| VLM query frequency $K$ | 50 epochs |
| Number of queries $N$ per session | 50 |
| Maximum budget of queries | 1500 |

*Table 4.* Hyperparameters for IQL in ALFRED.

We use a batch size of 512 and the Adam optimizer with learning rate 0.0001.

| Parameter | Value |
|---|---|
| Batch Size | 256 |
| # Training Steps | $1e5$ |
| Learning Rate | 0.0001 |
| Optimizer | Adam |
| Discount $\gamma$ | 0.99 |
| Q Update Polyak Averaging Coefficient | 0.005 |
| IQL Advantage Clipping | [0, 100] |
| IQL Advantage Inverse Temperature $\beta$ | 3 |
| IQL Qunatile $\tau$ | 0.7 |

*Table 5.* Hyperparameters for IQL in Real Robot.

## C. Baselines

We rerun RL-VLM-F using its original implementation[2], ensuring a fair comparison by using the same Gemini-1.5-Pro (Reid et al., 2024) as the VLM. While we can achieve similar reported performance in *Sweep Into* and *Open Drawer*, we observe lower performance in *Soccer*. For CLIP Score, we also use the implementation from RL-VLM-F, and we extend RoboCLIP Score based on the RL-VLM-F framework. Additionally, we observe that both CLIP and RoboCLIP achieve higher performance than previously reported in RL-VLM-F. In MetaWorld, task descriptions for CLIP, RoboCLIP, and RL-VLM-F remain the same as those used in (Wang et al., 2024a), as shown in Tables 11, 12, and 13. For ALFRED, we use task instructions as task description for all methods. The prompt template used in RL-VLM-F is shown in Table 10.

## D. Prompts and Task Descriptions

The prompt templates used to obtain ratings in ERL-VLM are shown in Table 6 (for MetaWorld), Table 7 (for ALFRED), and Table 8. The prompt templates for obtaining preferences in RL-VLM-F are provided in Table 9 (for MetaWorld) and Table 10 (for ALFRED).

---

[2]https://github.com/yufeiwang63/RL-VLM-F

---

**Prompt Template for ERL-VLM**

**Analyzing:**

`<Image>`

You will be presented an image of a robot arm performing the task `<Task Description>`. Please focus on the target object in the task and carefully analyze the image in term of completing the task.

- - - - - - - - - - - - - - - - - - - - - - - - - - - - - - - - - - - - - - - - - - - - - - - - - - - - - - - - - - - - - - -

**Rating:**

`<Analyses from previous response>`

From the above analyses, based on this rating category: `<Rating Classes>`, how would you rate this image in terms of completing task `<Task Description>`?

---

*Table 6.* Prompt Template for ERL-VLM used in MetaWorld environment: Rating classes can be {Bad, Average, Good}.

---

**Prompt Template for ERL-VLM**

**Analyzing:**

`<Image>`

You will be presented with an image containing a segment of the trajectory of a robot performing the task `<Instruction>`.
The trajectory segment contains `<N>` time steps of visual observations, corresponding to `<N-1>` intermediate actions.
The intermediate actions are: $<Action_1,...,Action_{N-1}>$.
Note that when an object is picked up, it appears close to the bottom of the view.

Please analyze the visual differences between consecutive time steps, reply the changes between consecutive time steps in each line explicitly. For example:
- Timestep 0 to 1 (Executed action): Your analysis
- Timestep 1 to 2 (Executed action): Your analysis
- and so on
The task is to `<Instruction>`, analyze this segment in terms of completing the task.

- - - - - - - - - - - - - - - - - - - - - - - - - - - - - - - - - - - - - - - - - - - - - - - - - - - - - - - - - - - - - - -

**Rating:**

`<Analyses from previous response>`

You are tasked with rating the RL agent's performance in completing a task `<Instruction>`. From the above analyses for `<N - 1>` transitions, based on this rating category: `<Bad, Average, Good>`, where:
- Bad: The task is not completed.
- Average: The task is partially completed, or you believe the transition is critical for achieving the goal, such as the target object appear in the agent's view, but falls short of completion.
- Good: The task is completed.
How would you rate each transition in terms of completing task?
Please reply a single line of list of ratings for `<N - 1>` transitions.
(For example: [rating of transition 1, rating of transition 2, ..., rating of transition `<N - 1>`).

---

*Table 7.* Prompt Template for ERL-VLM used in ALFRED environment.

---

**Prompt Template for ERL-VLM**

**Analyzing:**

`<Image>`

You will be presented an image of a robot arm performing the task `<Task Description>`. Please focus on the target object in the task and carefully analyze the image in term of achieving the task.

- - - - - - - - - - - - - - - - - - - - - - - - - - - - - - - - - - - - - - - - - - - - - - - - - - - - - - - - - - - - - - -

**Rating:**

`<Analyses from previous response>`

From the above analyses, based on this rating category: `<Rating Classes>`, how would you rate this image in terms of completing task `<Task Description>`?

---

*Table 8.* Prompt Template for ERL-VLM used in Real Robot environment.

---

**Prompt Template for RL-VLM-F**

**Analysis Template:**

Consider the following two images:
Image 1:
<Image 1>
Image 2:
<Image 2>
1. What is shown in Image 1?
2. What is shown in Image 2?
3. The goal is to <Task Description>. Is there any difference between Image 1 and Image 2 in terms of achieving the goal?

- - - - - - - - - - - - - - - - - - - - - - - - - - - - - - - - - - - - - - - - - - - - - - - - - - - - - - - - - - - - - -

**Labeling Template:**

Based on the text below to the questions:
[Repeat the 3 questions in the Analysis Template]
<VLM response>
Is the goal better achieved in Image 1 or Image 2? Reply a single line of 0 if the goal is better achieved in Image 1, or 1 if it is better achieved in Image 2.
Reply -1 if the text is unsure or there is no difference.

*Table 9.* Prompt Template for RL-VLM-F used in MetaWorld environment.

---

**Prompt Template for RL-VLM-F**

**Analysis Template:**

Consider the following two trajectory segments:
Trajectory Segment 1:
<Image 1>
Trajectory Segment 2:
<Image 2>
Trajectory segment 1 contains $<N_1>$ time steps of visual observations, corresponding to $<N_1 - 1>$ intermediate actions. The intermediate actions are: $<\texttt{Action}_1, \ldots, \texttt{Action}_{N_1-1}>$.
Trajectory segment 2 contains $<N_2>$ time steps of visual observations, corresponding to $<N_2 - 1>$ intermediate actions. The intermediate actions are: $<\texttt{Action}_1, \ldots, \texttt{Action}_{N_2-1}>$.
Note that when an object is picked up, it appears close to the bottom of the view.
Please analyze the visual differences between consecutive time steps of two trajectory segments, reply the changes between consecutive time steps in each line explicitly. For example:

Trajectory Segment 1:
- Timestep 0 to 1 (Executed action): Your analysis
- Timestep 1 to 2 (Executed action): Your analysis
- and so on

Trajectory Segment 2:
- Timestep 0 to 1 (Executed action): Your analysis
- Timestep 1 to 2 (Executed action): Your analysis
- and so on

The goal is <Instruction>. Is there any difference between Trajectory Segment 1 and Trajectory Segment 2 in terms of achieving the goal?

- - - - - - - - - - - - - - - - - - - - - - - - - - - - - - - - - - - - - - - - - - - - - - - - - - - - - - - - - - - - - -

**Labeling Template:**

Based on the text below to the questions:
<Analyses from previous response>
Is the goal better achieved in the Trajectory Segment 1 or Trajectory Segment 2?
Reply a single line of 0 if the goal is better achieved in Trajectory Segment 1, or 1 if it is better achieved in Trajectory Segment 2.
Reply -1 if the text is unsure of there is no difference.

*Table 10.* Prompt Template for RL-VLM-F used in ALFRED environment.

| Task Name | Task Description |
|---|---|
| *Sweep Into* | The green cube is in the hole. |
| *Drawer Open* | The drawer is opened. |
| *Soccer* | The soccer ball is in the goal. |

*Table 11.* Task descriptions for CLIP Score used in MetaWorld.

| Task Name | Task Description |
|---|---|
| *Sweep Into* | robot sweeping the green cube into the hole on the table |
| *Drawer Open* | robot opening green drawer |
| *Soccer* | robot pushing the soccer ball into the goal |

*Table 12.* Task descriptions for RoboCLIP used in MetaWorld.

| Task Name | Task Description |
|---|---|
| *Sweep Into* | to minimize the distance between the green cube and the hole |
| *Drawer Open* | to open the drawer |
| *Soccer* | to move the soccer ball into the goal |

*Table 13.* Task descriptions for RL-VLM-F used in MetaWorld.

| Task Name | Task Description |
|---|---|
| *Sweep Into* | place the green cube so that it lies on the square hole |
| *Drawer Open* | open the drawer |
| *Soccer* | place the soccer ball so that it lies inside the goal |

*Table 14.* Task descriptions for ERL-VLM used in MetaWorld.

| Task Name | Task Description |
|---|---|
| *Sweep Bowl* | Sweep the green bowl beyond the red line. |
| *Drawer Open* | Open the drawer with the blue handle until the green line is fully visible. |
| *Pickup Banana* | Pick up the banana on the gray plate. |

*Table 15.* Task descriptions for ERL-VLM used in Real Robot environment.

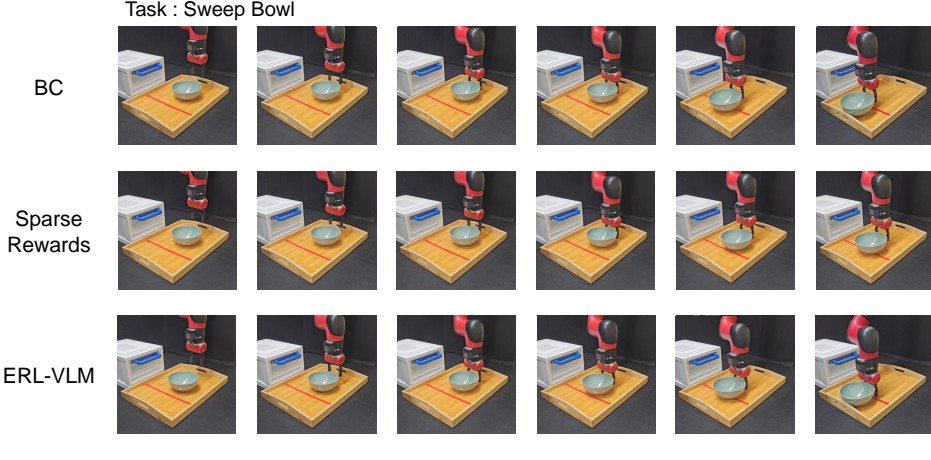

(a) The Sweep Bowl task is relatively simple, where the agent learns to push the bowl across the red line. As shown, all three methods achieve high success rates.

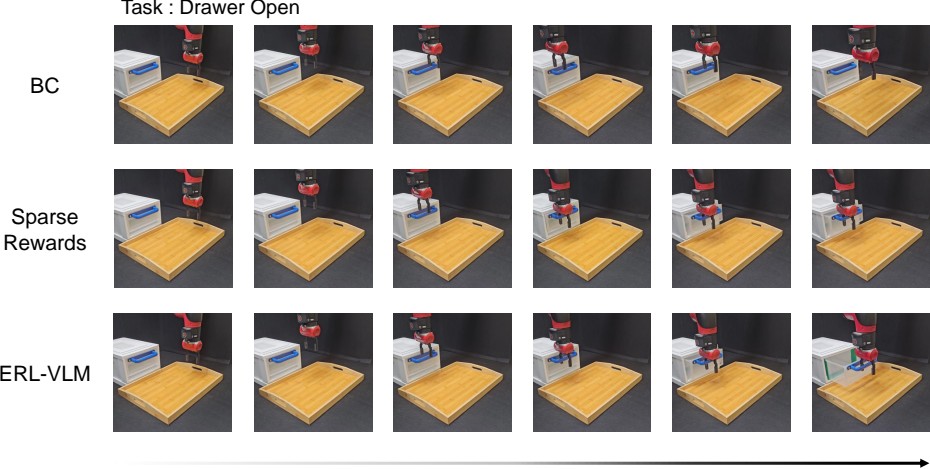

(b) In the Drawer Open task, both BC and Sparse Rewards failed to position the gripper correctly on the handle. In contrast, ERL-VLM successfully placed the gripper in the right spot and pulled the handle horizontally to complete the task.

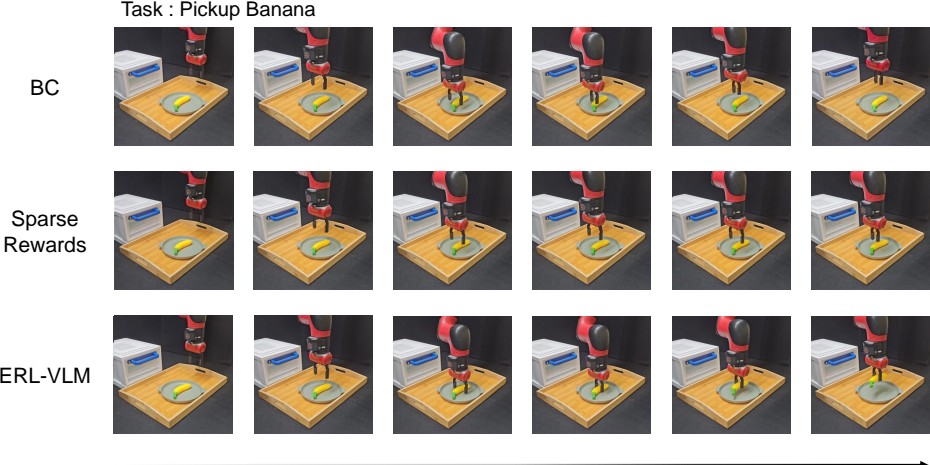

(c) In the Pickup Banana task, ERL-VLM successfully grasped and lifted the banana from the plate. In contrast, BC and Sparse Rewards moved close to the banana but failed to close the gripper at the correct position, resulting in lifting without proper grasping.

*Figure 9.* Visualizations of policy rollouts on three tasks in the real robot environment.

# E. Impact of MAE loss in Preference-based RL

We further investigate the impact of MAE loss in handling corrupted labels in RL-VLM-F (VLM-based PbRL) by replacing Cross-Entropy with MAE loss. The results, shown in Figure 10, indicate that MAE provides a slight improvement. This suggests that the lower performance may stems more from the ambiguity of preference queries rather than noisy labels.

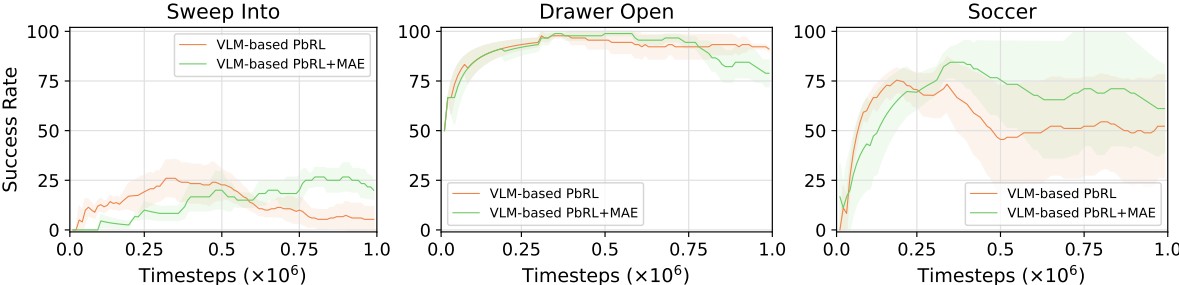

Figure 10. The impact of MAE on handling corrupted labels in RL-VLM-F.

# F. Additional Results & Analysis

## F.1. Distribution of Ratings

The rating distribution for the tasks is shown in Figure 11. As shown, for the three MetaWorld tasks, the percentage of Good ratings increases as the timestep progresses. For ALFRED tasks, the percentage of Average ratings increases over time, while the percentage of Good ratings remains almost constant. This is because, in ALFRED tasks, successful states occur at the end of the trajectory, so the VLM tends to assign Good ratings mostly at the final step, resulting in an increase in number of Good ratings, although the increase is relatively small compared to other ratings.

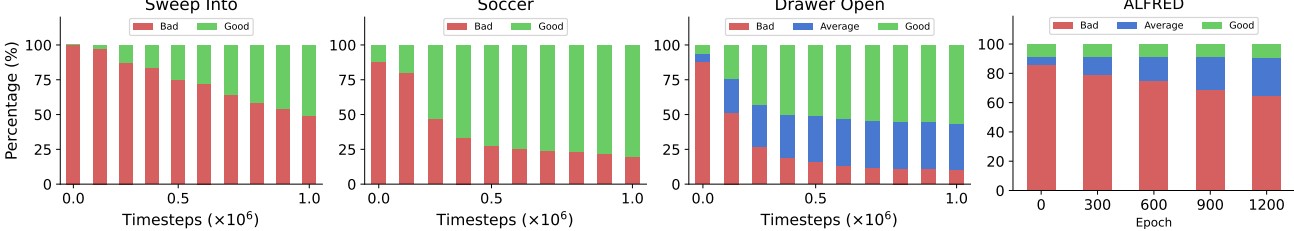

Figure 11. Typical rating distribution over the course of training for ERL-VLM agents. The first three plots show the distribution of ratings in MetaWorld tasks, while the final plot displays the distribution in ALFRED tasks, aggregated across four different task types.

## F.2. Examples of Segments Labeled with Different Ratings

We provide examples of states in MetaWorld and trajectories in ALFRED, each labeled with different ratings, in Figures 15, 16, 17, and 18. The reasoning behind the VLM's ratings (obtained during the analysis stage of our prompts) is also illustrated in these figures. As shown, the VLM assigns appropriate ratings based on the status of the target object in relation to the task objectives specified in the task description.

## F.3. Visualization of the Learned Reward

We visualize the outputs of the learned reward models from ERL-VLM, RL-VLM-F, and CLIP along expert trajectories in three MetaWorld tasks and four ALFRED tasks, as shown in Figures 19 and 20. In MetaWorld tasks, CLIP rewards are generally noisy and poorly aligned with task progress. While both ERL-VLM and RL-VLM-F exhibit increasing reward trends along expert trajectories, ERL-VLM aligns more closely with the ground-truth task progress and shows significantly less noise compared to RL-VLM-F. In ALFRED, ERL-VLM produces smoother and more consistent reward signals along expert trajectories than the other methods.

## F.4. Consistency of Ratings

In practice, we observe that the large VLMs (*e.g.*, Gemini) can produce stochastic ratings; identical segments may receive different labels, even when the temperature is set to zero. To investigate the consistency and reliability of these ratings, we

repeatedly query the VLM along expert trajectories across different tasks. The results, shown in Figures 12 and 13, illustrate the degree of rating consistency. While some noise is present, the overall ratings along expert trajectories remain stable and reasonable. For example, by calculating the accuracy of ratings based on task success/failure (for $n = 3$ rating levels, we only count Bad and Good), we observe the following rating accuracies: for Sweep Into, $90 \pm 4.6$; for Drawer Open, $65 \pm 3.8$; and for Soccer, $66.8 \pm 3.7$. The low variance across trials suggests that the inconsistencies are minor and can be effectively addressed by our MAE-based objective.

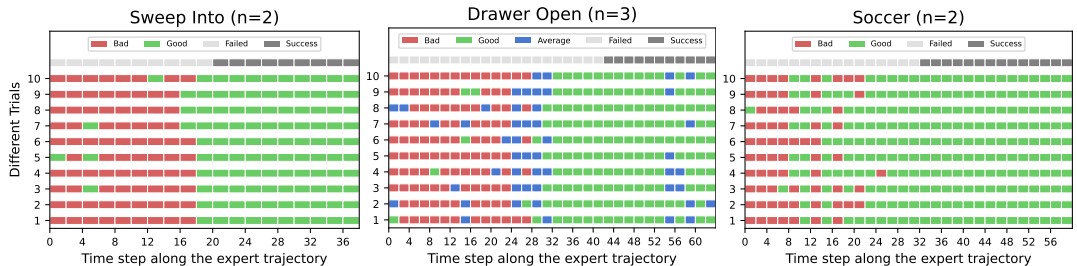

*Figure 12.* Consistency of VLM-assigned ratings for MetaWorld tasks: Using the expert trajectories from Figure 19, we perform 10 repeated trials to query the VLM for ratings (using the prompt template from Table 6, with the VLM temperature set to 0). The top row indicates task-specific success or failure at each timestep. The bottom 10 rows show the ratings from these trials along the expert trajectories (with every other timestep skipped for better visualization). The number of ratings, $n$, matches the value used in Figure 4.

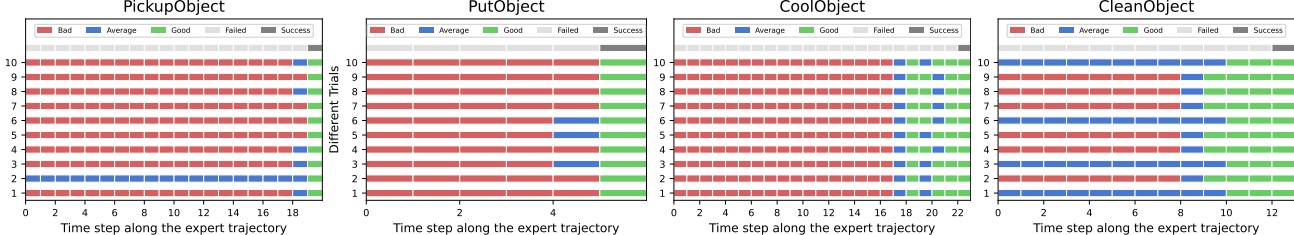

*Figure 13.* Consistency of VLM-assigned ratings for ALFRED tasks: Using the expert trajectories from Figure 20, we perform 10 repeated trials to query the VLM for ratings (using the prompt template from Table 7, with the VLM temperature set to 0). The top row indicates task-specific success or failure at each timestep. The bottom 10 rows show the ratings of trials along the expert trajectories. The number of ratings is equal to three across tasks.

The influence of rating levels on the consistency of VLM feedback in the Drawer Open task is further examined in Figure 14. With $n = 3$ ratings levels, the VLM produces more consistent and intuitive feedback, with ratings generally increasing as the trajectory progresses toward task completion. However, with other values (*e.g.*, $n = 2$ or $n = 4$), the VLM exhibits greater inconsistency. For instance, with $n = 2$, the VLM sometimes assigns Bad ratings even when the state reflects successful task completion. Similarly, with $n = 4$, it occasionally assigns Very Bad or Bad ratings to successful states. We hypothesize that this behavior stems from inherent biases in the VLM. Based on these observations, we recommend performing a simple consistency check when designing prompts for new tasks to guide the selection of an appropriate number of rating levels.

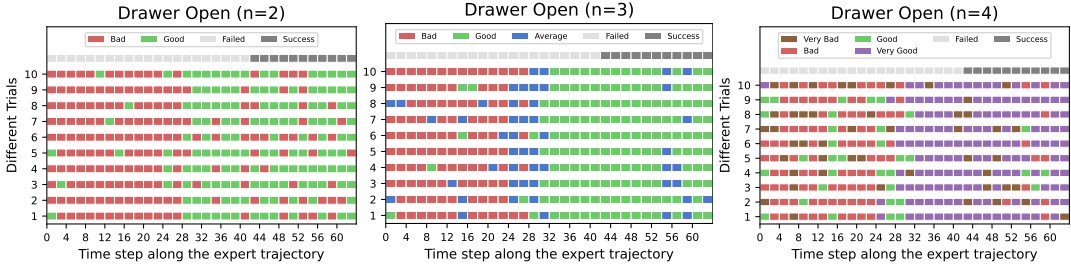

*Figure 14.* VLM-assigned ratings for the Drawer Open task with $n = \{2, 3, 4\}$. Using the expert trajectory from Figure 19, we perform 10 repeated trials to query the VLM for ratings (using the prompt template from Table 6, with the VLM temperature set to 0). The top row indicates task-specific success or failure at each timestep. The bottom 10 rows show the ratings from these trials along the expert trajectories (with every other timestep skipped for better visualization).

**Task:** Sweep Into

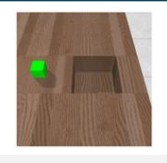

Prompt in Table 6

✦ **Gemini**

**Analyzing**: The image shows the green cube is not in the square hole. The task is therefore incomplete. The robot arm needs to move the cube and position it correctly over and within the hole to complete the task.
**Rating**: Bad

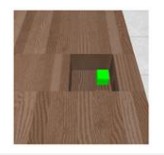

Prompt in Table 6

✦ **Gemini**

**Analyzing**: The image shows the green cube already placed within the square hole. Therefore, the task of "place the green cube so that it lies on the square hole" is complete.
**Rating**: Good

**Task:** Drawer Open

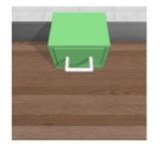

Prompt in Table 6

✦ **Gemini**

**Analyzing**: The image shows a small, single-drawer cabinet with the drawer closed. The task is to open the drawer. Therefore, the task is not completed in this image. The robot arm would need to grasp the handle and pull it forward to complete the task.
**Rating**: Bad

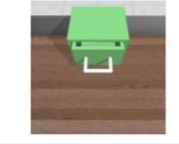

Prompt in Table 6

✦ **Gemini**

**Analyzing**: The drawer is already slightly open, so the task is partially completed. To fully complete the task of opening the drawer, the robot arm would need to grasp the handle and pull it towards itself until the drawer is fully extended.
**Rating**: Average

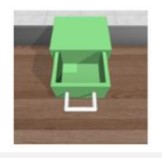

Prompt in Table 6

✦ **Gemini**

**Analyzing**: The image shows the drawer already in an open state. Therefore, the task of opening the drawer is already completed.
**Rating**: Good

**Task:** Soccer

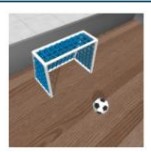

Prompt in Table 6

✦ **Gemini**

**Analyzing**: The image shows the soccer ball outside and in front of the goal. The task is not complete. The ball needs to be moved into the space between the posts and behind the net in order for it to be considered "inside the goal.
**Rating**: Bad

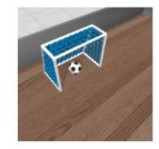

Prompt in Table 6

✦ **Gemini**

**Analyzing**: The image shows the soccer ball already inside the goal. Therefore, the task of placing the soccer ball inside the goal is complete.
**Rating**: Good

*Figure 15.* Examples of states with different ratings in MetaWorld tasks. We use the prompt template from Table 6 to obtain ratings. The images shown were used to query the VLM, and the text outputs are directly taken from the model's response. Following (Wang et al., 2024a), we remove the robot from the images to facilitate the VLM's analysis.

**Task Instruction:** "Pick up the mug in the coffee maker"

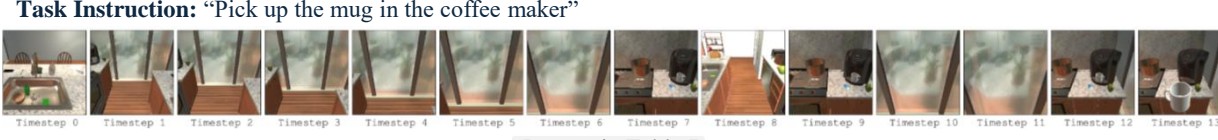

Prompt in Table 7

✦ **Gemini**

**Analyzing**:
- Timestep 0 to 1 (RotateRight): The robot rotates to the right, the view changes from the front of the counter to the side of the counter.
- Timestep 1 to 2 (MoveAhead): The robot moves forward, the view shows more of the counter and the coffee maker.
- Timestep 2 to 3 (MoveAhead): The robot moves forward, the view shows more of the counter and the coffee maker.
- Timestep 3 to 4 (MoveAhead): The robot moves forward, the view shows more of the counter and the coffee maker.
- Timestep 4 to 5 (MoveAhead): The robot moves forward, the view shows more of the counter and the coffee maker.
- Timestep 5 to 6 (MoveAhead): The robot moves forward, the view shows more of the counter and the coffee maker.
- Timestep 6 to 7 (MoveAhead): The robot moves forward, the view shows more of the counter and the coffee maker.
- Timestep 7 to 8 (MoveAhead): The robot moves forward, the view shows more of the counter and the coffee maker.
- Timestep 8 to 9 (MoveAhead): The robot moves forward, the view shows more of the counter and the coffee maker.
- Timestep 9 to 10 (RotateLeft): The robot rotates to the left, the view changes from the side of the counter to the front of the counter.
- Timestep 10 to 11 (LookDown): The robot looks down, the view changes from the front of the counter to a closer view of the coffee maker.
- Timestep 11 to 12 (PickupObject): The robot picks up the mug, the mug is now close to the bottom of the view.

**Rating**: ["Bad","Bad","Bad","Bad","Bad","Bad","Bad","Bad","Bad","Bad","Average","Good"]

---

**Task Instruction:** "Pick up the mug in the coffee maker"

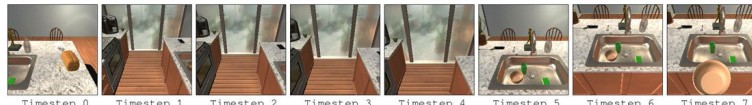

Prompt in Table 7

✦ **Gemini**

**Analyzing**:
- Timestep 0 to 1 (RotateLeft): The agent turns left from facing the sink to facing the window.
- Timestep 1 to 2 (MoveAhead): The agent moves closer to the window.
- Timestep 2 to 3 (MoveAhead): The agent moves closer to the window.
- Timestep 3 to 4 (MoveAhead): The agent moves closer to the window.
- Timestep 4 to 5 (MoveAhead): The agent moves closer to the window.
- Timestep 5 to 6 (MoveAhead): The agent moves closer to the window.
- Timestep 6 to 7 (RotateLeft): The agent turns left from facing the window to facing the coffee maker.
- Timestep 7 to 8 (RotateLeft): The agent turns left from facing the coffee maker to facing the kitchen island.
- Timestep 8 to 9 (RotateRight): The agent turns right from facing the kitchen island to facing the coffee maker.
- Timestep 9 to 10 (RotateRight): The agent turns right from facing the coffee maker to facing the window.
- Timestep 10 to 11 (MoveAhead): The agent moves closer to the window.
- Timestep 11 to 12 (RotateLeft): The agent turns left from facing the window to facing the coffee maker.
- Timestep 12 to 13 (PickupObject): The agent picks up the mug from the coffee maker.

**Rating**: ["Bad", "Bad", "Bad", "Bad", "Bad", "Bad", "Bad", "Bad", "Bad", "Bad", "Bad", "Bad", "Good"]

---

**Task Instruction:** "Pick up the mug in the coffee maker"

Prompt in Table 7

✦ **Gemini**

**Analyzing**:
- Timestep 0 to 1 (RotateLeft): The agent rotates to the left, bringing the counter into view. The bread is no longer visible.
- Timestep 1 to 2 (MoveAhead): The agent moves closer to the counter. The coffee maker and the mug are larger in the view.
- Timestep 2 to 3 (MoveAhead): The agent moves closer to the counter. The coffee maker and mug are even larger, and the apple is now visible.
- Timestep 3 to 4 (MoveAhead): The agent moves closer to the counter, now very close to the coffee maker. The mug is now the primary object in view.
- Timestep 4 to 5 (RotateRight): The agent rotates to the right, bringing the sink into view. The mug is still visible, but the coffee maker is mostly out of view.
- Timestep 5 to 6 (LookDown): The agent looks down. The sink is closer in the view, and the mug is no longer visible.
- Timestep 6 to 7 (PickupObject): The agent picks up the bowl from the sink. The bowl is now large and at the bottom of the view.

**Rating**: ["Average", "Average", "Average", "Average", "Bad", "Bad", "Bad"]

*Figure 16.* Examples of trajectories with different ratings in a PickObject task in ALFRED. We use the prompt template from Table 7 to obtain ratings. The images shown were used to query the VLM, and the textual outputs are taken directly from the model's responses.

**Task Instruction:** "Pick up the tomato from the sink"

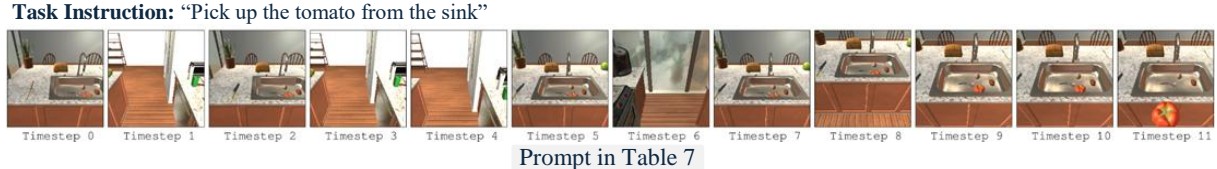

Prompt in Table 7

✦ **Gemini**

**Analyzing:**
- Timestep 0 to 1 (RotateRight): The robot rotates to the right, the view changes from the counter to the side of the kitchen.
- Timestep 1 to 2 (MoveAhead): The robot moves forward, the view changes to a closer view of the stove.
- Timestep 2 to 3 (MoveAhead): The robot moves forward, the view changes to a closer view of the stove.
- Timestep 3 to 4 (MoveAhead): The robot moves forward, the view changes to a closer view of the stove.
- Timestep 4 to 5 (MoveAhead): The robot moves forward, the view changes to a closer view of the stove.
- Timestep 5 to 6 (RotateRight): The robot rotates to the right, the view changes from the stove to the counter.
- Timestep 6 to 7 (RotateRight): The robot rotates to the right, the view changes from the counter to the side of the kitchen.
- Timestep 7 to 8 (MoveAhead): The robot moves forward, the view changes to a closer view of the counter.
- Timestep 8 to 9 (RotateLeft): The robot rotates to the left, the view changes from the side of the kitchen to the counter.
- Timestep 9 to 10 (MoveAhead): The robot moves forward, the view changes to a closer view of the sink.
- Timestep 10 to 11 (RotateRight): The robot rotates to the right, the view changes from the sink to the side of the kitchen.
- Timestep 11 to 12 (MoveAhead): The robot moves forward, the view changes to a closer view of the counter.
- Timestep 12 to 13 (RotateLeft): The robot rotates to the left, the view changes from the side of the kitchen to the counter.
- Timestep 13 to 14 (LookDown): The robot looks down, the view changes to a closer view of the sink.
- Timestep 14 to 15 (PickupObject): The robot picks up the tomato, the tomato appears close to the bottom of the view.

**Rating:** ["Bad", "Bad", "Bad", "Bad", "Bad", "Bad", "Bad", "Bad", "Bad", "Bad", "Bad", "Bad", "Bad", "Average", "Good"]

---

**Task Instruction:** "Pick up the tomato from the sink"

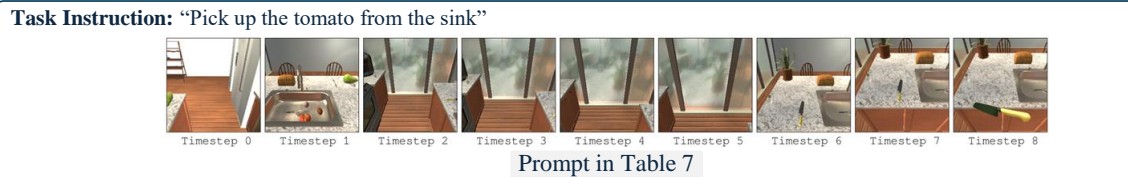

Prompt in Table 7

✦ **Gemini**

**Analyzing:**
- Timestep 0 to 1 (RotateRight): The view rotates slightly to the right, now facing more of the floor and less of the counter.
- Timestep 1 to 2 (RotateLeft): The view rotates back to the left, returning to roughly the same position as in timestep 0.
- Timestep 2 to 3 (RotateRight): The view rotates to the right again, similar to timestep 1.
- Timestep 3 to 4 (MoveAhead): The view moves forward, now showing a wider view of the kitchen, including a ladder and part of the fridge.
- Timestep 4 to 5 (RotateLeft): The view rotates to the left, now facing the counter with the sink.
- Timestep 5 to 6 (RotateLeft): The view rotates further to the left, now showing the stove and a window.
- Timestep 6 to 7 (RotateRight): The view rotates back to the right, now facing the sink again.
- Timestep 7 to 8 (LookDown): The view tilts downwards, focusing on the sink and its contents.
- Timestep 8 to 9 (MoveAhead): The view moves slightly closer to the sink.
- Timestep 9 to 10 (MoveAhead): The view moves even closer to the sink.
- Timestep 10 to 11 (PickupObject): The tomato, which was previously in the sink, is now at the bottom of the view, indicating it has been picked up.

**Rating:** ["Bad", "Bad", "Bad", "Bad", "Bad", "Bad", "Bad", "Bad", "Bad", "Average", "Good"]

---

**Task Instruction:** "Pick up the tomato from the sink"

Prompt in Table 7

✦ **Gemini**

**Analyzing:**
- Timestep 0 to 1 (RotateLeft): The agent rotates to the left, now facing the sink more directly. The tomato and potato are more centered in the view.
- Timestep 1 to 2 (RotateLeft): The agent rotates further left, now facing away from the sink and towards the cabinets. The sink is partially out of view.
- Timestep 2 to 3 (MoveAhead): The agent moves forward, closer to the cabinets. The sink is no longer visible.
- Timestep 3 to 4 (MoveAhead): The agent continues moving forward, still facing the cabinets.
- Timestep 4 to 5 (MoveAhead): The agent moves forward again, now close to the cabinets.
- Timestep 5 to 6 (RotateRight): The agent rotates right, now facing the sink and counter again. The knife is visible on the counter.
- Timestep 6 to 7 (LookDown): The agent looks down, bringing the knife closer to the bottom of the view. The sink is more visible.
- Timestep 7 to 8 (PickupObject): The agent picks up the knife. It is now large and at the bottom of the view.

**Rating:** ["Bad", "Bad", "Bad", "Bad", "Bad", "Bad", "Bad", "Bad"]

*Figure 17.* Examples of trajectories with different ratings in a PickObject task in ALFRED. We use the prompt template from Table 7 to obtain ratings. The images shown were used to query the VLM, and the textual outputs are taken directly from the model's responses.

**Task Instruction:** "Open the fridge door, put the tomato inside of the fridge, close the door, open the door, take the tomato out, close the door."

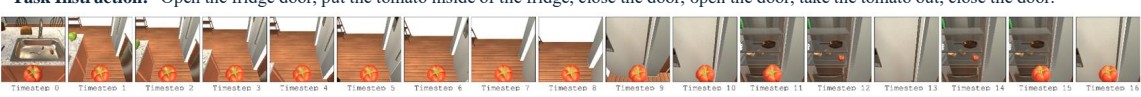

Prompt in Table 7

◆ **Gemini**

**Analyzing**:
- Timestep 0 to 1 (RotateRight): The robot rotates to the right, the view changes from the kitchen counter to the side of the fridge.
- Timestep 1 to 2 (MoveAhead): The robot moves forward, the view shows more of the fridge.
- Timestep 2 to 3 (MoveAhead): The robot moves forward, the view shows more of the fridge.
- Timestep 3 to 4 (MoveAhead): The robot moves forward, the view shows more of the fridge.
- Timestep 4 to 5 (MoveAhead): The robot moves forward, the view shows more of the fridge.
- Timestep 5 to 6 (MoveAhead): The robot moves forward, the view shows more of the fridge.
- Timestep 6 to 7 (MoveAhead): The robot moves forward, the view shows more of the fridge.
- Timestep 7 to 8 (MoveAhead): The robot moves forward, the view shows more of the fridge.
- Timestep 8 to 9 (RotateRight): The robot rotates to the right, the view changes to the front of the fridge.
- Timestep 9 to 10 (OpenObject): The robot opens the fridge door, the view shows the inside of the fridge.
- Timestep 10 to 11 (PutObject): The robot puts the tomato inside the fridge, the tomato is now on the top shelf of the fridge.
- Timestep 11 to 12 (CloseObject): The robot closes the fridge door, the view shows the front of the fridge.
- Timestep 12 to 13 (OpenObject): The robot opens the fridge door, the view shows the inside of the fridge.
- Timestep 13 to 14 (PickupObject): The robot picks up the tomato, the tomato is now close to the bottom of the view.
- Timestep 14 to 15 (CloseObject): The robot closes the fridge door, the view shows the front of the fridge.

**Rating**: ["Bad", "Bad", "Bad", "Bad", "Bad", "Bad", "Bad", "Bad", "Bad", "Average", "Good", "Good", "Average", "Good", "Good"]

---

**Task Instruction:** "Open the fridge door, put the tomato inside of the fridge, close the door, open the door, take the tomato out, close the door."

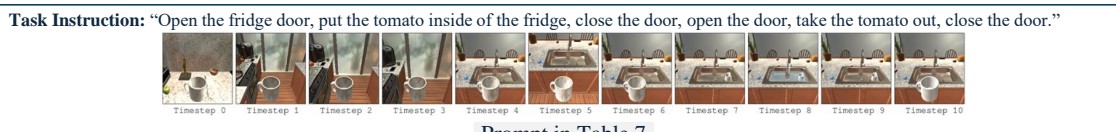

Prompt in Table 7

◆ **Gemini**

**Analyzing**:
- Timestep 0 to 1 (RotateRight): The view rotates to the right, the tomato is stationary.
- Timestep 1 to 2 (MoveAhead): The view moves forward slightly, the tomato is stationary.
- Timestep 2 to 3 (MoveAhead): The view moves forward, the tomato is stationary.
- Timestep 3 to 4 (MoveAhead): The view moves forward, the tomato is stationary.
- Timestep 4 to 5 (MoveAhead): The view moves forward, the tomato is stationary.
- Timestep 5 to 6 (MoveAhead): The view moves forward, the tomato is stationary.
- Timestep 6 to 7 (MoveAhead): The view moves forward, the tomato is stationary.
- Timestep 7 to 8 (MoveAhead): The view moves forward, the tomato is stationary.
- Timestep 8 to 9 (RotateRight): The view rotates to the right, the tomato is stationary.
- Timestep 9 to 10 (LookUp): The view looks up, the tomato is stationary.
- Timestep 10 to 11 (OpenObject): The fridge door opens, the tomato is stationary.
- Timestep 11 to 12 (PutObject): The tomato is placed inside the fridge.
- Timestep 12 to 13 (CloseObject): The fridge door is closed.
- Timestep 13 to 14 (OpenObject): The fridge door is opened.
- Timestep 14 to 15 (PickupObject): The tomato is picked up.
- Timestep 15 to 16 (CloseObject): The fridge door is closed.

**Rating**: ["Bad", "Bad", "Bad", "Bad", "Bad", "Bad", "Bad", "Bad", "Bad", "Bad", "Average", "Good", "Good", "Average", "Good", "Good"]

---

**Task Instruction:** "Open the fridge door, put the tomato inside of the fridge, close the door, open the door, take the tomato out, close the door."

Prompt in Table 7

◆ **Gemini**

**Analyzing**:
- Timestep 0 to 1 (RotateRight): The agent rotates to the right, the view changes from the counter with a mug, apple, golden statue, and bowl to a view facing the stove and fridge.
- Timestep 1 to 2 (MoveAhead): The agent moves closer to the stove and fridge. The mug is now closer to the camera.
- Timestep 2 to 3 (MoveAhead): The agent moves closer to the stove and fridge. The mug is even closer to the camera.
- Timestep 3 to 4 (RotateRight): The agent rotates right, now facing the sink. The mug is still held.
- Timestep 4 to 5 (LookDown): The agent looks down, providing a better view of the mug and the sink.
- Timestep 5 to 6 (LookUp): The agent looks up, providing a wider view of the sink area.
- Timestep 6 to 7 (PutObject): The agent places the mug in the sink.
- Timestep 7 to 8 (ToggleObjectOn): The faucet is turned on, filling the sink with water.
- Timestep 8 to 9 (ToggleObjectOff): The faucet is turned off, the water stops filling the sink.
- Timestep 9 to 10 (PickupObject): The agent picks up the mug from the sink. It is now close to the bottom of the view.

**Rating**: ["Bad","Bad","Bad","Bad","Bad","Bad","Bad","Bad","Bad","Bad"]

*Figure 18.* Examples of trajectories with different ratings in a CoolObject task in ALFRED. We use the prompt template from Table 7 to obtain ratings. The images shown were used to query the VLM, and the textual outputs are taken directly from the model's responses.

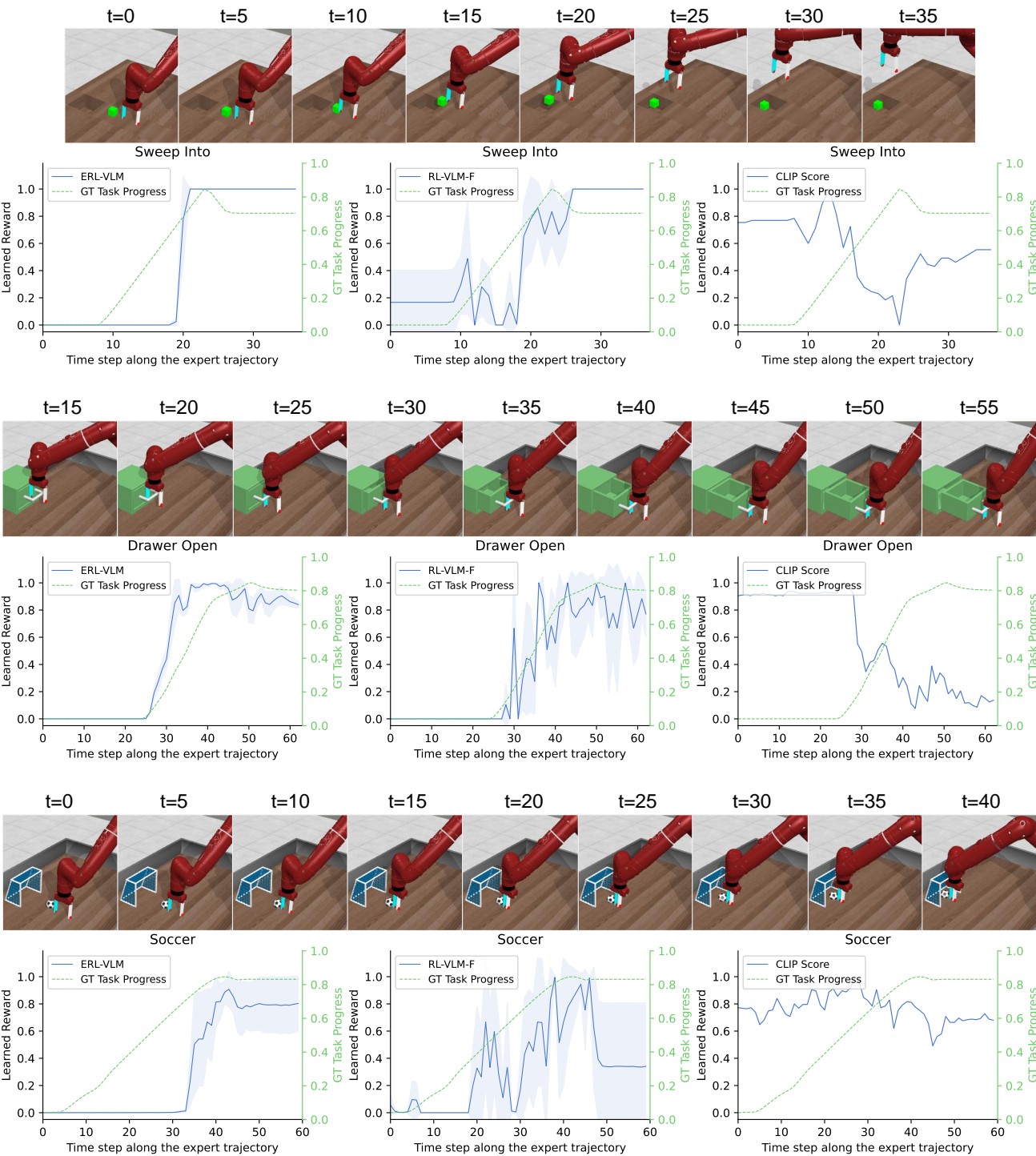

*Figure 19.* Comparison of reward outputs from ERL-VLM, RL-VLM-F, and CLIP against ground-truth task progress in MetaWorld tasks. The ground-truth task progress is computed following (Wang et al., 2024a): for Sweep Into, it is measured as the negative distance between the cube and the hole; for Drawer Open, as the distance the drawer has been pulled out; and for Soccer, as the negative distance between the soccer ball and the goal. We normalize both the reward values and ground-truth task progress into the range of [0, 1] for a better comparison between them. The learned rewards are averaged over three runs, with shaded regions indicating standard deviation. Images are rendered at corresponding timesteps from the expert trajectories.

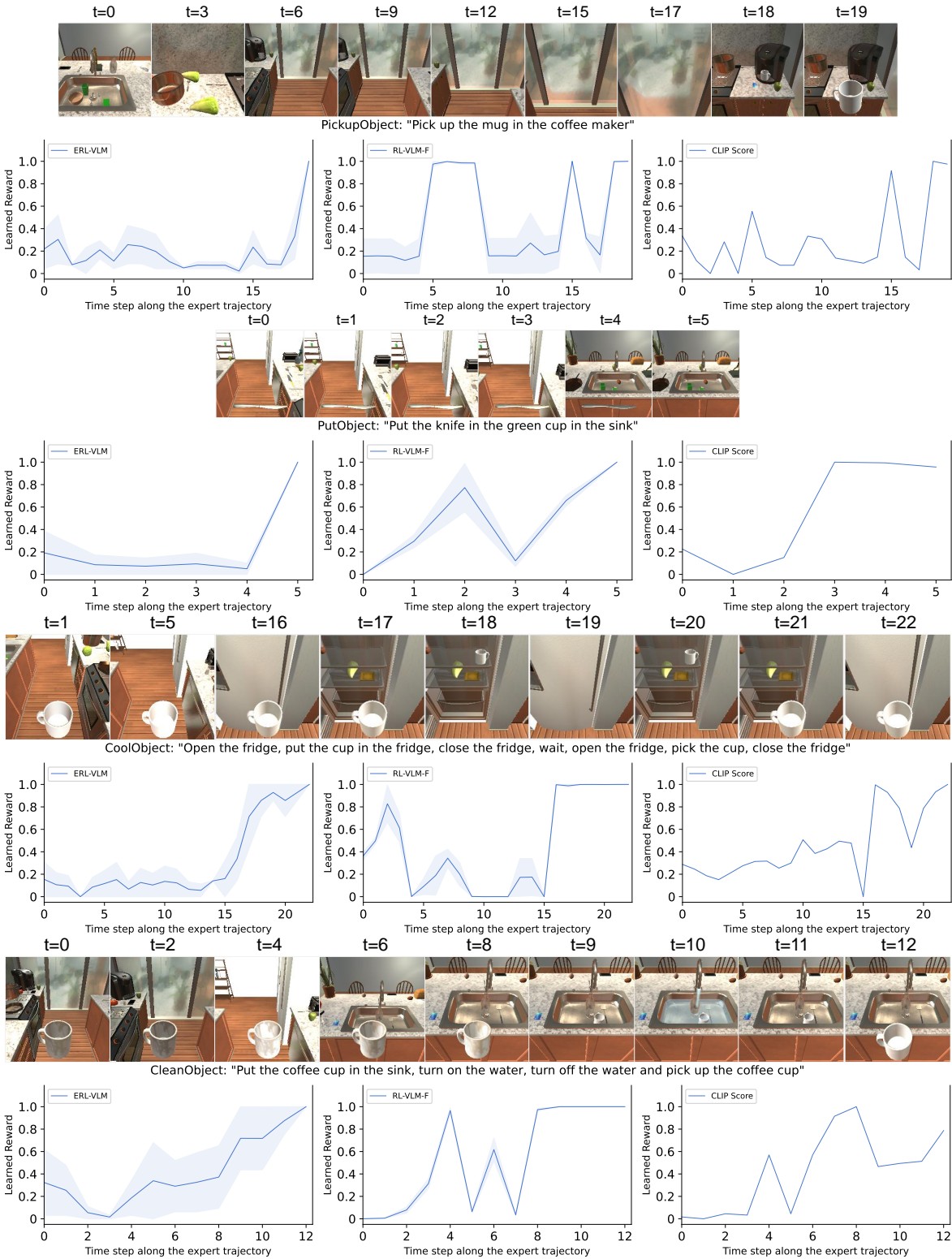

*Figure 20.* Comparison of reward outputs from ERL-VLM, RL-VLM-F, and CLIP across four ALFRED tasks along expert trajectories. Ground-truth task progress is omitted as it is not available in ALFRED. We normalize the reward values into the range of [0, 1] for a better comparison. Images are rendered at corresponding timesteps from the expert trajectories.

