# OpenReview forum: "Enhancing Rating-Based Reinforcement Learning to Effectively Leverage Feedback from Large Vision-Language Models"
_ICML.cc/2025/Conference — ICML 2025 poster_

### Official Review · Reviewer_p9Em · 2025-03-11

**Overall Recommendation:** 4

**Summary:**

This paper introduces ERL-VLM, a straightforward yet effective method for learning reward functions by leveraging feedback from VLMs. By querying a VLM for absolute ratings of trajectory segments rather than relying on pairwise comparisons, the method aims to improve sample efficiency and expressiveness of the feedback signal. The authors augment this approach with stratified sampling and a mean absolute error (MAE) training objective to address issues related to data imbalance and noisy labels. Experimental results, spanning both simulated environments and real-world robotic tasks, demonstrate that ERL-VLM outperforms four baseline methods in six out of seven tasks, thereby examining the effectiveness of ERL-VLM.

**Claims And Evidence:**

The claims are generally well supported by empirical results, as the improvements achieved over baselines are notable. However, some technical aspects, such as the details of the trajectory segment sampling and the validity of VLM ratings, could benefit from further clarification. In particular, it is unclear whether all sampled segments are informative, since early or mid-trajectory segments might not contain sufficient task-relevant information, and whether the observed teacher VLM accuracy (approximately 0.7 as shown in Figure 2(a)) adequately justifies its use.

**Essential References Not Discussed:**

N/A

**Experimental Designs Or Analyses:**

- The authors conduct experiments across different environments and tasks. Success rate is used as the main metric to evaluate the quality of the reward function, which is sound.

- As aforementioned, one concern is about the quality of the labels generated by the VLM. Since the output from generative models can be subjective and unstable due to the input prompt, namely, an identical segment may obtain different labels.

- The authors conduct experiments in a real robot, which is commendable.

**Methods And Evaluation Criteria:**

The proposed method leverages a VLM for trajectory rating and two enhancement techniques to mitigate the intrinsic issues from the VLM (e.g., imbalance and imprecision). The workflow mainly resolves around how to enhance the feedback from the VLM, thereby optimizing the reward function. The major concerns from my end are about the segment sampling and the validity of the ratings generated by the VLM.

While the approach is sound, there are two primary concerns:

- Trajectory Segment Sampling: The use of random sampling without consideration of temporal dynamics might yield segments that do not adequately reflect the task’s critical moments. From Algorithm 1 Line 12, trajectory segments are randomly sampled from the replay buffer and fed to the VLM to acquire labels/rating. It is not clear whether the sampled segments are qualified to provide valid ratings. Namely, some segments from the beginning of the operation may not contain useful information for rating even from humans. Also, some tasks can be determined as good/succeed or bad/failed only at the very end of the operation, therefore, it is hard to justify a meaningful rating for the trajectory in the middle. It is not convincing that a random sampling approach without considering the temporal dependence and importance of the segments can collect trajectories efficiently and effectively.

- VLM Rating Reliability: With a teacher VLM accuracy hovering around 0.7, further quantitative analysis is needed to explore the consistency and informativeness of these ratings across different scenarios.

**Other Comments Or Suggestions:**

- Some of the technical details in the Appendix regarding reward learning could be integrated into the main body to provide a clearer technical overview.

- Although the experiments use only Gemini-1.5 Pro as the VLM, the choice is justified given its representative performance, and the authors acknowledge that extending to other VLMs is a potential future direction.

**Other Strengths And Weaknesses:**

- The authors may justify more about using the Likert scale for rating, since, to the best of my knowledge, both floating numbers and Likert scale suffer a degree of uncertainty for evaluating/labelling.

-  In section 5, the authors discuss the impact of the number of rating classes. From Figure 6, it seems the selection of the number of rating classes is somehow task-specific. The authors may elabrate more about this point.

**Questions For Authors:**

1. Could you elaborate on the criteria for selecting trajectory segments from the replay buffer? How do you ensure these segments provide sufficient context for accurate ratings, especially in cases where critical task information might appear only at the end of the trajectory?

2. Can you provide additional quantitative analysis on the consistency and reliability of the ratings across different tasks and scenarios?

3. What is the rationale behind choosing a Likert scale for the ratings, given that both discrete scales and continuous ratings can suffer from inherent uncertainties?

**Relation To Broader Scientific Literature:**

The related work covers relevant methods and some of them are applied as baselines for comparison. It is worth noting that some works [1-3] use LLMs or VLMs as coding models to directly generate reward functions without computing similarity score. The authors may add some discussion regarding these approaches.

[1] Eureka: Human-Level Reward Design via Coding Large Language Models

[2] Self-refined large language model as automated reward function designer for deep reinforcement learning in robotics

[3] AutoReward: Closed-Loop Reward Design with Large Language Models for Autonomous Driving

**Theoretical Claims:**

This paper does not include theoretical proof, and the main contribution is not in such an aspect.

---

> ### Author Rebuttal · Authors · 2025-04-01
>
> We thank the reviewer for their in-depth review, and for recognizing the simplicity and effectiveness of our method, the strength of our results, and our efforts in real-world experiments. We address your comments in detail below:
> - **Q1**. Could you elaborate on the criteria for selecting trajectory segments from the replay buffer?
>   - **A1**. For MetaWorld, we randomly sample segments of length one (i.e., individual states) and use them to query the VLM. This approach is sufficient for the single-target nature of the tasks, as shown in Figure 1 at [Link](https://drive.google.com/file/d/19DtB_xCUhg8uTmTbE5bPDwXFoEn71PkL/view?usp=sharing). For ALFRED, since tasks are often compositional and critical information may appear at different points in the episode, we randomly sample an entire trajectory and use it to query the VLM to provide sufficient context for accurate ratings, as illustrated in Figures 2-4 at [Link](https://drive.google.com/file/d/19DtB_xCUhg8uTmTbE5bPDwXFoEn71PkL/view?usp=sharing).
> - **Q2**. Can you provide additional quantitative analysis on the consistency and reliability of the ratings across different tasks and scenarios?
>   - **A2**. The consistency of the VLM ratings is shown in Figures 9 and 10 at [Link](https://drive.google.com/file/d/19DtB_xCUhg8uTmTbE5bPDwXFoEn71PkL/view?usp=sharing). We repeatedly query the VLM along expert trajectories in different tasks. As pointed out by the reviewer, we also observe that identical segments may receive different labels. However, as shown in the results, the ratings along expert trajectories remain reasonable and consistent. For example, by calculating the accuracy of ratings based on task success/failure (for $n=3$ rating levels, we only count *Bad* and *Good*), we observe the following rating accuracies: for *Sweep Into*, $90\pm4.6$; for *Drawer Open*, $65 \pm 3.8$; and for *Soccer*, $66.8 \pm 3.7$. The variance across trials is small, and these small inconsistencies can be mitigated through our MAE objective.
> - **Q3**. What is the rationale behind choosing a Likert scale for the ratings?
>   - **A3**. The rationale for selecting a Likert scale is as follows:
>   1. Human alignment: Since our reward labels are generated by a VLM designed to emulate human interpretation, a Likert scale aligns well with common human rating systems. This allows the VLM to map qualitative assessments (e.g., *Bad*, *Average*, *Good*) to structured numerical values, which would be more challenging to express meaningfully using an arbitrary continuous scale.
>   2. Stability and reliability: It is considerably more stable and reliable to prompt VLMs with structured classification tasks (e.g., “Rate the success of this grasp from 1 to 5”) than to request continuous scalar (e.g., “Give a score between 0 and 1”). Continuous scores often yield unstable or inconsistent outputs, as shown in prior work [1], whereas discrete classes reduce ambiguity, providing clearer grounding for ratings.
>   3. Scalability and prior work: Prior research in learning from evaluative feedback [2-5] often discretizes qualitative feedback into ordinal scales for reward modeling. These pipelines report that such formats are easier to scale, annotate, and integrate into training loops.
> - **Q4**: Discussion with some works use LLMs or VLMs as coding models to directly generate reward functions without computing similarity score.
>   - **A4**. The relevant works suggested by the reviewer utilize LLMs/VLMs as coding models that directly generate reward functions by accessing environment source code or privileged state information. In contrast, our method relies solely on visual observations and does not require access to internal environment representations. We will incorporate the works recommended by the reviewer ([6-8]) into our revised manuscript to enrich the related work section and more clearly position our method within the broader literature.
>
> [1] Wang, Y. et al. RL-VLM-F: Reinforcement Learning from Vision-Language Foundation Model Feedback, ICML 2024.
>
> [2] Knox, W.B. et al. Interactively shaping agents via human reinforcement: The tamer framework. 2019.
>
> [3] Warnell, G. et al. Deep tamer: Interactive agent shaping in highdimensional state spaces. 2018.
>
> [4] Arumugam, D. et al. Deep reinforcement learning from policy-dependent human feedback. 2019.
>
> [5] White, D. et al. Rating-based Reinforcement Learning, AAAI 2024.
>
> [6] Wu, Y. et al. Eureka: Human-Level Reward Design via Coding Large Language Models.
>
> [7] Li, X. et al. Self-refined Large Language Model as Automated Reward Function Designer for Deep Reinforcement Learning in Robotics.
>
> [8] Chen, C. et al. AutoReward: Closed-Loop Reward Design with Large Language Models for Autonomous Driving.

---

> > ### Comment · Reviewer_p9Em · 2025-04-02
> >
> > Thanks for the additional details and analysis, which have addressed my concerns. Hence, I would like to raise my score to Accept.

---

> > > ### Author Response · Authors · 2025-04-02
> > >
> > > Thank you for your prompt response and for raising the score!
> > >
> > > We are glad that our response has addressed your concerns. We greatly appreciate the time and effort you spent providing insightful feedback on our work.

---

### Official Review · Reviewer_94ge · 2025-03-13

**Overall Recommendation:** 4

**Summary:**

The paper introduces ERL-VLM, a method that efficiently utilizes feedback from large VLMs like Gemini to generate reward functions for training RL agents. Instead of pairwise comparisons, it queries VLMs for absolute evaluations of individual trajectories on a Likert scale. This approach allows for more expressive feedback, reduces ambiguity, and ensures full utilization of queried samples. The authors also introduce enhancements to existing rating-based RL methods to address instability caused by data imbalance and noisy rating labels from VLMs.  Through extensive experiments across various vision-language navigation and robotic control tasks, ERL-VLM is shown to outperform prior VLM-based reward generation methods. Ablation studies are conducted to identify key performance factors and provide insights into its effectiveness.

**Claims And Evidence:**

Yes

**Essential References Not Discussed:**

No

**Experimental Designs Or Analyses:**

Yes. This paper conducts a series of experiments to evaluate the ERL-VLM method. In the simulation domain, it is compared with various baselines, showing obvious advantages in most tasks. Real-robot experiments demonstrate that this method can generate effective reward functions to facilitate policy learning. Ablation experiments analyze the contributions of various improvements to performance, revealing that the MAE loss, stratified sampling, etc., are effective, while label smoothing is ineffective. Experiments on different numbers of rating classes reveal the relationship between task characteristics and the optimal number of rating classes.

**Methods And Evaluation Criteria:**

Yes

**Other Comments Or Suggestions:**

No

**Other Strengths And Weaknesses:**

### Strengths
- A novel method, ERL-VLM, for learning reward functions from vision - language model feedback is proposed, providing new ideas and approaches for the design of reward functions in reinforcement learning.
- Experiments are conducted not only on various types of tasks in simulated environments but also in real-world robot operation scenarios.
- Detailed ablation experiments are carried out on each improved part of the method, clearly demonstrating the impact of each improvement measure on the overall performance.

### Weaknesses
- The method is highly dependent on the performance and accuracy of vision-language models. What if the vision-language model consistently exhibits understanding biases and inaccurate ratings in some complex scenarios?
- An accuracy metric for the reward function should be introduced to measure the degree of matching between the learned reward function and the real task objective. In environments with sparse rewards, what specific reward designs does ERL-VLM derive that lead to performance improvements?
- ERL-VLM seems to rely on the task description. How should it handle multiple target tasks?

**Questions For Authors:**

No

**Relation To Broader Scientific Literature:**

1. Addressing the Challenge of Reward Design.
2. Leveraging VLM Capabilities.
3. Overcoming VLM-based Feedback Challenges.

**Theoretical Claims:**

No

---

> ### Author Rebuttal · Authors · 2025-04-01
>
> We thank the reviewer for the detailed and thoughtful review, and for recognizing the novelty of our method, the extensiveness of our experiments, and the strength of our results across a wide range of tasks and domains. We respond to your comments in detail below:
>
> - **Q1**. The method is highly dependent on the performance and accuracy of vision-language models. What if the vision-language model consistently exhibits understanding biases and inaccurate ratings in some complex scenarios?
>
>   - **A1**. We acknowledge that biases and inaccuracies in VLM ratings can arise, particularly in complex scenarios. To mitigate these issues, we focus on designing effective prompts and selecting appropriate rating levels, which help guide the VLM’s output. Additionally, using higher-performing VLMs can further enhance the quality of the ratings. However, if a VLM consistently produces inaccurate ratings due to being undertrained on specific tasks or domains, the finetuning for the VLMs is required.
> - **Q2**. An accuracy metric for the reward function should be introduced to measure the degree of matching between the learned reward function and the real task objective. In environments with sparse rewards, what specific reward designs does ERL-VLM derive that lead to performance improvements?
>
>   - **A2**. ERL-VLM performs better in sparse reward tasks because the reward model, learned from ratings, provides denser reward signals at meaningful states, rather than only sparse signals at the final state.
>
>     To measure the alignment between the learned reward function and the real task objective, we compare the reward outputs from ERL-VLM with the ground-truth task progress along expert trajectories. The results are shown in Figures 7 and 8 at [Link](https://drive.google.com/file/d/19DtB_xCUhg8uTmTbE5bPDwXFoEn71PkL/view?usp=sharing) . Outputs from RL-VLM-F and CLIP are also included for comparison. As shown, ERL-VLM rewards are more consistent and less noisy compared to the other methods. In sparse reward tasks, our approach enables reward shaping by assigning higher values to meaningful states. For instance, in the *CoolObject* task (Figure 8), ERL-VLM assigns higher rewards at timesteps when the agent interacts with the target object (e.g., the fridge), compared to intermediate navigation steps. This behavior arises from the VLM assigning higher ratings to critical subgoal states, as illustrated in Figures 2–4 at [Link](https://drive.google.com/file/d/19DtB_xCUhg8uTmTbE5bPDwXFoEn71PkL/view?usp=sharing).
> - **Q3**. ERL-VLM seems to rely on the task description. How should it handle multiple target tasks?
>
>   - **A3**: ERL-VLM handles multiple target tasks (i.e., compositional task descriptions) by querying the VLM with a sequence of images rather than a single image, as illustrated in Figures 2-4 at [Link](https://drive.google.com/file/d/19DtB_xCUhg8uTmTbE5bPDwXFoEn71PkL/view?usp=sharing). We find that incorporating temporal context helps the VLM better understand the overall task structure, leading to more accurate ratings in compositional task descriptions.

---

### Official Review · Reviewer_Rx7z · 2025-03-16

**Overall Recommendation:** 4

**Summary:**

This paper studies the problem of automated reward generation for RL policy training via VLMs. While prior work has shown that rewards can be extracted from VLMs either by preference (relative comparison) between two or more trajectory segments, or by using the representation itself as a distance metric, these approaches are often prone to instability (former) or local minima (latter). This paper proposes a framework for reward generation via VLMs based on a rating system (e.g. "bad", "ok", "good") rather than direct trajectory comparison, which (a) allows for more fine-grained quantitative feedback, and (b) circumvents the issue of training instability when sampled trajectories are highly similar. The VLM assigns ratings to trajectory segments (one or multiple image frames), which are then used to learn a reward model for downstream RL training. The authors find that naively training a reward model on the collected dataset leads to poor predictions due to its skewed data distribution (most early trajectories are expected to be bad in an online RL setting), and instead propose to use a stratified sampling as well as an MAE objective. Experiments are conducted on 3 tasks from Meta-World, as well as ALFRED. Results indicate that the proposed framework is effective at guiding online RL via learned rewards on the selected tasks, outperforming simpler baselines based on representation distance or pairwise preferences.

## Post-rebuttal assessment

I appreciate all the new qualitative analysis of the method provided in the rebuttal, and believe that it addresses my concerns. I have raised my score from Weak Accept -> Accept under the assumption that these new results will be included in the camera-ready version.

**Claims And Evidence:**

The main claims of this paper are (1) that their framework, ERL-VLM, is that it enables agents to learn new tasks using only a human-provided language task description, (2) that their proposed improvements to rating-based reward learning improve stability and thereby agent performance, (3) that their framework outperforms prior work on VLM-based reward generation, and finally (4) that their ablations identify key factors of their framework for good performance. I believe that all of these claims are reasonable and for the most part supported by convincing evidence. However, while it is true that their ablations identify key factors for performance, I find the analysis of *why* somewhat lacking; I go into more detail on this in the "experimental designs and analysis" section.

**Essential References Not Discussed:**

I believe that the discussion of related work is fairly comprehensive and I am not aware of any essential references that are not currently mentioned. It is however possible that I might have missed some.

**Experimental Designs Or Analyses:**

The experimental design appears fairly sound. While the number of simulated tasks is rather small, the authors validate their method across a combination of continuous (Meta-World) and multi-task language-conditioned discrete (ALFRED) control, as well as three simple manipulation tasks on a real robot. Experiments are conducted with 3 seeds; while additional seeds would make the experiments more sound, performance gains appear to be statistically significant as is. Both the VLM, reward model, and policy take RGB images as input which makes the framework broadly applicable.

My main concern with the paper in its current form is the lack of analysis on what exactly trajectories labelled different ratings look like, what the typical distribution of ratings is for the considered tasks, and what the reward model outputs for successful vs. unsuccessful trajectories. It would be informative to analyze this from both a quantitative and a more qualitative perspective. Based on the ablation on number of ratings in Figure 6, it appears to me that the ratings might actually correspond to task stages (e.g. reaching, grasping, and pulling for the Drawer Open task). This would explain why the number of ratings has a pretty drastic effect on policy learning, since many tabletop manipulation tasks like the ones studied here can be divided quite naturally into a small number of distinct stages. It feels important to analyze this in more detail, as it will provide insights into the rating-based framework proposed in the paper while it would also help inform readers how to select the number of ratings (task stages?) if applied to novel problems.

**Methods And Evaluation Criteria:**

Yes. The improvements to rating-based reward generation are well motivated and backed by empirical evidence.

**Other Comments Or Suggestions:**

It would be helpful to more explicitly mention that both VLMs, reward model, and policy takes RGB images as input. I had to read through the appendices to confirm that the policy was also image-based.

**Other Strengths And Weaknesses:**

**Strengths:** The paper is very well written and easy to follow. The discussion of related work appears to be thorough, and the experiments are fairly comprehensive and insightful. The problem is timely and likely to be of interest to the community.

**Weaknesses:** There is limited quantitative and qualitative analysis wrt what exactly the ratings represent in the context of the chosen tasks; I believe that including some insights in this regard would greatly strengthen the paper. The technical content of the paper is somewhat incremental, so more analysis, insights, and takeaways would likely improve longevity and impact of the work.

**Questions For Authors:**

I would like the authors to address my comments in the "experimental design and analysis" and "weaknesses" sections above, especially my comments regarding the effect of # ratings, what the ratings and rewards look like in practice (quantitatively and qualitatively), and how they relate to task stages.

**Relation To Broader Scientific Literature:**

The contributions of this paper, while somewhat incremental, are well motivated wrt prior work. The authors motivate their approach by providing insights into the limitations of current approaches for VLM-based reward generation, which is backed by empirical results.

**Theoretical Claims:**

N/A

---

> ### Author Rebuttal · Authors · 2025-04-01
>
> We thank the reviewer for their insightful and constructive feedback, and for recognizing the clear motivation behind ERL-VLM, the clarity of our writing, and the strength of our experimental results. We address your comments in detail below:
>
> - **Q1**. What do trajectories labeled with different ratings look like in practice?
>
>   - **A1**. Sixteen examples of states/trajectories labeled with different ratings are shown as follows:
>
>     - MetaWorld tasks (*Sweep Into*, *Drawer Open*, and *Soccer*), where each task includes two or three different states, each leading to different ratings assigned by the VLM. These examples are shown in Figure 1 at [Link](https://drive.google.com/file/d/19DtB_xCUhg8uTmTbE5bPDwXFoEn71PkL/view?usp=sharing).
>
>     - ALFRED tasks with three different instructions, each paired with three trajectories leading to varying ratings. These are shown in Figures 2, 3, and 4 at [Link](https://drive.google.com/file/d/19DtB_xCUhg8uTmTbE5bPDwXFoEn71PkL/view?usp=sharing).
>
>
>    The reasoning behind the VLM’s ratings (obtained during the analysis stage of our prompts) is also illustrated in the figures. As shown, the VLM assigns appropriate ratings based on the status of the target object in relation to the task objectives specified in the task description.
>
> - **Q2**. What is the typical distribution of ratings for the considered tasks?
>
>   - **A2**. The typical rating distribution for the tasks is shown in Figure 5 at [Link](https://drive.google.com/file/d/19DtB_xCUhg8uTmTbE5bPDwXFoEn71PkL/view?usp=sharing). As shown, for the first three MetaWorld tasks, the percentage of *Good* ratings increases as the timestep progresses. For ALFRED tasks, the percentage of *Average* ratings increases over time, while the percentage of *Good* ratings remains almost constant. This is because, in ALFRED tasks, successful states occur at the end of the trajectory, so the VLM tends to assign *Good* ratings mostly at the final step, resulting in an increase in number of *Good* ratings, although the increase is relatively small compared to other ratings.
> - **Q3**. What does the output of the reward model look like in practice?
>
>   - **A3**. The outputs of the reward models from ERL-VLM along expert trajectories in three MetaWorld tasks and four ALFRED tasks are shown in Figures 7 and 8 at [Link](https://drive.google.com/file/d/19DtB_xCUhg8uTmTbE5bPDwXFoEn71PkL/view?usp=sharing). We also include outputs from RL-VLM-F and CLIP for comparison. In MetaWorld tasks, CLIP rewards are generally noisy and poorly aligned with task progress. While both ERL-VLM and RL-VLM-F exhibit increasing reward trends along expert trajectories, ERL-VLM aligns more closely with the ground-truth task progress and shows significantly less noise compared to RL-VLM-F. In ALFRED, ERL-VLM produces smoother and more consistent reward signals along expert trajectories than the other methods.
> - **Q4**. The effect of #ratings.
>
>   - **A4**. As mentioned in **Q1**, the VLM assigns ratings based on the status of the target object. For many tasks, the appropriate number of ratings can be easily determined (e.g., whether the task has been successfully completed is clearly observable from the visual state). However, for tasks where determining the optimal number of ratings is non-trivial, we use a trial-and-error approach. We demonstrate this process in Figure 6 at [Link](https://drive.google.com/file/d/19DtB_xCUhg8uTmTbE5bPDwXFoEn71PkL/view?usp=sharing), where we repeatedly query the VLM along an expert trajectory in the *Drawer Open* task.
>
>     With $n=3$ ratings levels, the VLM produces more consistent and intuitive feedback, with ratings generally increasing as the trajectory progresses toward task completion. Small inconsistencies are mitigated through our MAE objective. However, with other values (e.g., $n=2$ or $n=4$), the VLM exhibits stronger inconsistency. For example, with n=2, the VLM sometimes assigns *Bad* ratings even when the state reflects successful task completion. Similarly, with n=4, both *Very Bad* and *Bad* ratings are occasionally assigned to successful states. We hypothesize that this behavior stems from inherent biases in the VLM. Based on this observation, we recommend performing a simple consistency check when designing prompts for new tasks to help select an appropriate number of rating levels.

---

### Decision · Program_Chairs · 2025-05-01

**Decision:**

Accept (poster)

**Comment:**

This work introduces a novel and effective method for learning reward functions in reinforcement learning (RL) using absolute ratings from large VLMs. The authors demonstrate ERL-VLM's superiority over existing VLM-based reward generation methods through comprehensive experiments across simulated (Meta-World, ALFRED) and real-world robotic tasks, achieving significant performance improvements in six out of seven tasks. The proposed enhancements, including stratified sampling and MAE loss, address critical issues like data imbalance and noisy labels, as shown in ablation studies. The rebuttal effectively addresses reviewer concerns, particularly regarding trajectory segment sampling and VLM rating reliability. The work is a meaningful contribution to the field.